# OpenReview forum: "Counterfactual Time Series Forecasting with Textual Conditions"
_ICLR.cc/2026/Conference — Submitted to ICLR 2026_

### Official Review · Reviewer_FZhN · 2025-10-27

**Soundness:** 3
**Presentation:** 4
**Contribution:** 3
**Rating:** 4
**Confidence:** 4

**Summary:**

This paper introduces TADIFF, a method for counterfactual time series prediction, conditioned on textual information about the future. This diffusion-based approach learns to separate the intrinsic time series properties from the conditioning signals, allowing realistic future predictions. They propose a counterfactual fine-tuning using synthetic counterfactual samples that improves prediction performance on unseen conditions. The paper also proposes 2 evaluation metrics (DTTC-I and DTTC-E) for evaluating counterfactual instances with no underlying ground truth, which is a common challenge in counterfactual forecasting.

**Strengths:**

1) The problem motivation and setup are clearly explained, supported by examples and clear figures.

2) The idea of using an intrinsic feature of the time-series sequence as the initial seed and then adding text conditioning is interesting and intuitive. As noted later, additional ablation tests could further strengthen this argument.

3) I believe the paper is well motivated, and the ability to make counterfactual time-series predictions using future text conditions is valuable and can be utilized in many applications.

**Weaknesses:**

1) The motivation for creating a synthetic counterfactual dataset is clear, but the current version does not justify whether the synthetic samples are reasonable or realistic under real conditions. For instance, a) Is randomly sampling future conditions realistic? In many real-world settings, future conditions depend on prior ones; wouldn’t this yield unrealistic sequences of text conditions?
b) You select the condition closest to $c_h$, but this choice is not justified. Shouldn’t it be closest to $c_f$? Why should sampled text conditions be close to historical conditions? Overall, synthetic datasets need more thorough evaluation.

2) Add more detail about the generative process of the synthetic dataset. How are the text conditions generated? In the examples provided in the Appendix, the texts seem tightly tied to intrinsic time-series features (e.g., “trend going up”). But shouldn't the text conditions provide complementary information? This distinction is important to justify decoupling time-series–specific features from the conditioning signal. If the two modalities are so intertwined, is the decoupling meaningful?

3) Related to the 2 points mentioned above, a well-designed synthetic dataset would let you properly evaluate the synthetic data generation process. With full control over the generation process (including counterfactuals), you can assess the quality of the counterfactual dataset, show it is realistic, and demonstrate that it generates samples from counterfactual conditions not present in the data. I think this might be a good way to justify the CF dataset quality.

4) The benefit of fine-tuning with the counterfactual dataset is not fully convincing. Performance drops with counterfactual fine-tuning on the Traffic dataset (vs. without), which is undesirable and suggests potential biases in the CF data. Does this pattern repeat on other datasets? If so, the fine-tuning approach may need to be reconsidered. Gains on counterfactual data are not worth losses elsewhere, especially when the curated dataset is not validated for realism. Also authors can evaluate how fine-tuning affects benchmark performance as well, as a general solution.

5) “Finding 8” claims robustness to hyperparameters, but the figure does not clearly support this, metrics especially vary substantially on ETTM1. Can the authors explain why this is considered robust? Also, can you provide a justification for the reverse relationship between DTTC-E and DTTC-I.

**Questions:**

1) The paper mentions that initializing inference with Gaussian random noise may introduce properties that conflict with the condition. Can you please justify this claim and give an example? Why would this conflict arise? Intuitively, such noise might instead conflict with intrinsic features of the historical sequence. Please clarify, and ideally include an ablation comparing your proposed initialization to random-noise initialization.

2) In forecasting, is concatenating  $x_{h,0}$ and $x_{h,T}$ necessary? Doesn’t  $x_{h,T}$ already encode all past information?

3) How is the conditioning signal $c$ extracted from text? If a specific embedding model is used, please provide details for reproducibility. Also, how sensitive is overall performance to the quality of this embedding model?

4) “Finding 2” on page 8 cites the wrong table(s). Please correct the reference.

5) This is more of a suggestion, the related work section should also cover causal inference fundamentals: what guarantees the field provides and how this work is situated among existing approaches.

---

> ### Author Response · Authors · 2025-11-21
> **Rebuttal by Authors (part 1 of 5)**
>
> > **W1.1** The motivation for creating a synthetic counterfactual dataset is clear, but the current version does not justify whether the synthetic samples are reasonable or realistic under real conditions. For instance, a) Is randomly sampling future conditions realistic? In many real-world settings, future conditions depend on prior ones; wouldn’t this yield unrealistic sequences of text conditions?
>
> Thanks for your questions. We agree that it is important to justify that the constructed counterfactual samples are reasonable.
>
> For the real-world datasets, we construct counterfactual data through random sampling + similarity filtering to ensure that it is as reasonable as possible. Our goal is to create future conditions that differ from the historical patterns while avoiding significant conflicts. As detailed in Sec.3.5, the diversity is achieved by randomly sampling future conditions from factual data, and the plausibility is ensured through a similarity-based filtering step.
>
> We also conduct an experiment to evaluate the quality of the constructed counterfactual data.
> - We compare the similarity between historical and future conditions across factual data and constructed counterfactual data. Two construction strategies are considered: (a) random sampling combined with similarity filtering, which is the method we adopt, and (b) random sampling only.
> - The results are shown in Table 1 below, where "factual" refers to the original factual data, "counterfactual" refers to the data constructed using our method, and "random" represents data generated through random sampling alone. The results indicate that our constructed counterfactual data exhibit higher similarity to factual data compared with the purely random samples. This demonstrates that the counterfactual data produced by our method are reasonable and share a distribution closer to the factual data.
>
> #### Table 1. Similarity analysis between factual and two types of counterfactual data.
>
> |Datasets|$\text{Sim}\_{\text{factual}}$|$\text{Sim}\_{\text{counterfactual}}$|$\text{Sim}\_{\text{random}}$|
> |:--:|:--:|:--:|:--:|
> |ETTm1|165.59|155.96|104.83|
> |Exchange|294.85|287.68|110.63|
> |Traffic|86.12|84.47|72.64|
> |Weather|55.57|57.42|46.05|
>
> > **W1.2** b) You select the condition closest to $c\_h$, but this choice is not justified. Shouldn’t it be closest to $c\_f$? Why should sampled text conditions be close to historical conditions? Overall, synthetic datasets need more thorough evaluation.
>
> Thanks for your questions.
>
> We follow Leibniz's Principle of Continuity (cited in Sec.3.3 of our paper), which states that natural change is continuous and gradual. This is consistent with the reviewer’s view that “future conditions depend on prior ones.” Therefore, counterfactual conditions should not exhibit significant conflicts with the historical pattern $c\_h$.
>
> In other words, selecting the condition closest to $c\_f$ can also be considered reasonable under the guidance of Leibniz's Principle of Continuity. We believe that the fundamental principle of constructing counterfactual data is to ensure diversity while maintaining the plausibility of the counterfactual conditions, that is avoiding strong conflicts with historically observed patterns, as discussed in Sec.3.5 of our paper.
>
> > **W2.1** Add more detail about the generative process of the synthetic dataset. How are the text conditions generated?
>
> Thanks for your suggestions.
>
> The generative process of the synthetic dataset can be splited into the following steps:
> - Six types of attribute (Trend Type, Trend Direction, Seasonality Cycles, Local Shapelets, Noise, Bias) are defined. Each type of attribute is associated with heuristic function to construct the time series, as discussed in Appendix B.2.1.
> - The six types of attribute are divided into intrinsic features and external conditions, where Trend Type, Noise and Bias belong to intrinsic features; Trend Direction, Seasonality Cycles and Local Shapelets belong to external conditions. We have added the details about this in the latest version.
> - The attributes belonging to external conditions are transferred into the textual captions using diverse templates, as discussed in Appendix B.2.2.

---

> ### Author Response · Authors · 2025-11-21
> **Rebuttal by Authors (part 2 of 5)**
>
> > **W2.2** In the examples provided in the Appendix, the texts seem tightly tied to intrinsic time-series features (e.g., “trend going up”). But shouldn't the text conditions provide complementary information? This distinction is important to justify decoupling time-series–specific features from the conditioning signal. If the two modalities are so intertwined, is the decoupling meaningful?
>
> Thanks for your questions.
>
> In the synthetic dataset, Trend Direction is treated as an external condition rather than an intrinsic feature. In fact, all attributes that appear in the textual descriptions can be regarded as external conditions, since they may change in future textual annotations. or example, the historical caption may state “trend going up”, while the future caption may state “trend going down.” Attributes that do not appear in the text are treated as intrinsic features.
>
> The text conditions indeed provide complementary information. They describe how the time series will evolve in the future, information that is difficult to infer solely from historical observations. In the synthetic dataset, some attributes of the time series are influenced by the text while others are not, and we achieve this decoupling by assigning different roles to different attributes.
>
> The intertwinement between time series and text is common, even in real-world scenarios. If the text is relevant to how the time series evolves, such intertwinement naturally exists. For example, in traffic forecasting, extreme weather events can cause decreases in traffic flow. In our synthetic dataset, we simplify this intertwinement into more direct textual descriptions such as “trend going down”.
>
> > **W3** Related to the 2 points mentioned above, a well-designed synthetic dataset would let you properly evaluate the synthetic data generation process. With full control over the generation process (including counterfactuals), you can assess the quality of the counterfactual dataset, show it is realistic, and demonstrate that it generates samples from counterfactual conditions not present in the data. I think this might be a good way to justify the CF dataset quality.
>
> Thanks for your suggestions. We agree that a well-designed synthetic dataset enables more rigorous evaluation of counterfactual forecasting.
>
> We believe the synthetic dataset we use is fully controlled, as discussed in the response to **W2.1**. We further assess its quality to verify that the counterfactual conditions in the test split do not appear in the train split, i.e., one intrinsic property of counterfactual setting.
> - Experiment setting. We count the number of attribute combinations across different splits of the synthetic dataset to demonstrate that the evaluation indeed involves conditional combinations not observed during training. Note that our defined conditional combinations include not only combinations of different attributes but also combinations of historical and future attributes. Therefore, the theoretical upper bound of the number of combinations in the synthetic dataset is $(2(\text{trend direction})\times 4(\text{season cycle}) \times 4^3(\text{local shaplet in 3 areas}))^2=262,144$.
> - Experiment results. The statistics are reported in Table 2 below, where #Train and #Test denote the number of attribute combinations in the train and test splits, respectively. We observe that fewer than half of the combinations appear in both splits, while the others constitute entirely unseen conditions for evaluation. This confirms that the synthetic dataset provides reliable counterfactual scenarios and that our model (TADiff) is capable of generalizing to unseen conditions.
>
> #### Table 2. The number of attribute combination on different splits of Synth.
> ||#Train|#Test|#Train & Test|#Only Test|
> |:--:|:--:|:--:|:--:|:--:|
> |Number|6733|1150|498|652|

---

> ### Author Response · Authors · 2025-11-21
> **Rebuttal by Authors (part 3 of 5)**
>
> > **W4** The benefit of fine-tuning with the counterfactual dataset is not fully convincing. Performance drops with counterfactual fine-tuning on the Traffic dataset (vs. without), which is undesirable and suggests potential biases in the CF data. Does this pattern repeat on other datasets? If so, the fine-tuning approach may need to be reconsidered. Gains on counterfactual data are not worth losses elsewhere, especially when the curated dataset is not validated for realism. Also authors can evaluate how fine-tuning affects benchmark performance as well, as a general solution.
>
> Thanks for your suggestions.
>
> We believe that finetuning with the counterfactual dataset can improve the semantic consistency metrics (DTTC-I and DTTC-E) while maintaining the numerical accuracy metrics (MAE and MSE).
> - We conduct the ablation study by running each setting 3 times with different random seeds. The MAE/MSE differences between the finetuning and non–finetuning settings fall within the range of the standard deviation, indicating that the differences are not statistically significant.
> - We also conduct paired t-tests between the results of the two methods, and the resulting p-values are significantly higher than 0.05, further confirming that the differences are not statistically significant. Similar patterns are also observed on the ETTm1 dataset, as discussed in Finding 3.
>
> Based on the results in Finding 3, we conclude that:
> - DTTC scores are not in conflict with traditional forecasting metrics (MAE/MSE). To some extent, they measure related properties of forecasting quality.
> - Finetuning our model on counterfactual data using DTTC scores as the optimization objective is meaningful and beneficial. This procedure improves the DTTC scores without harming performance on MAE or MSE.
>
> > **W5** “Finding 8” claims robustness to hyperparameters, but the figure does not clearly support this, metrics especially vary substantially on ETTM1. Can the authors explain why this is considered robust? Also, can you provide a justification for the reverse relationship between DTTC-E and DTTC-I.
>
> Thanks for your questions.
>
> The robustness of the hyperparameters is discussed within the context of a reasonable range. The influence of the hyperparameters (loss weight ratios) remains stable in most cases, and significant performance differences are observed only when the ratios take extreme values. This pattern is particularly evident for the loss weight ratio between forecasting and attribution $\lambda\_F:\lambda\_A$, as shown in the left subfigure of Figure 5.
>
> The inverse relationship between DTTC-E and DTTC-I is intuitive and expected, since these two metrics are directly linked to their corresponding optimization objectives. Increasing the loss weight of one objective naturally improves the performance on its associated metric. Therefore, we adopt a balanced loss weight ratio to achieve more uniform performance across metrics, as discussed in Finding 8.
>
> > **Q1** The paper mentions that initializing inference with Gaussian random noise may introduce properties that conflict with the condition. Can you please justify this claim and give an example? Why would this conflict arise? Intuitively, such noise might instead conflict with intrinsic features of the historical sequence. Please clarify, and ideally include an ablation comparing your proposed initialization to random-noise initialization.
>
> Thanks for your questions.
>
> Your understanding that noise sampled from a standard Gaussian distribution may “conflict with the intrinsic features of the historical sequence” is correct. We apologize for the ambiguous wording in the last paragraph of Sec.3.2, and we have refined the statement in the latest version of the paper. Similar to the concept discussed in the response to **W2**, a time series consists of intrinsic features and external conditions. The attribution process aims to remove the external conditions described in the text while preserving the intrinsic features as much as possible.
>
> We also conduct experiments to demonstrate the existence of this conflict and analyze its impact on forecasting performance.
> - Obvious distribution difference between the our initialization and random Gaussian initialization. As discussed in Finding 5, the attribution-guided noise space forms more compact clusters than the random Gaussian noise in the attribution-free setting. The component missing compared with a random Gaussian distribution is precisely the noise that conflicts with the intrinsic features of the time series in the corresponding dataset.
> - The conflict between random Gaussian initialization and the intrinsic features of the historical sequence degrades forecasting performance. As discussed in Finding 4, text-attribution significantly improves the DTTC-I score, which evaluates whether the forecasts preserve the intrinsic features of the historical sequence.

---

> ### Author Response · Authors · 2025-11-21
> **Rebuttal by Authors (part 4 of 5)**
>
> > **Q2** In forecasting, is concatenating $x\_{h,0}$ and $x\_{h,T}$ necessary? Doesn’t $x\_{h,T}$ already encode all past information?
>
> Thanks for your questions.
>
> We believe that incorporating $x\_{h,0}$ during forecasting is still necessary. The statement that “$x\_{h,T}$ already encodes all past information” is not entirely accurate, but it does offer a useful perspective on how many intrinsic features can be preserved during the attribution process ($x\_{h,0}\rightarrow x\_{h,T}$). Since the attribution process is not fully controllable and $x\_{h,T}$ is essentially in a noise-like form, it becomes difficult to preserve certain types of information, such as numerical ranges.
>
> To verify this, we conduct an ablation study on the Synth dataset to examine the influence of $x\_{h,0}$. Specifically, we compare forecasting performance with and without $x\_{h,0}$. The results in Table 3 show that removing $x\_{h,0}$ still yields comparable semantic consistency (DTTC-E), while the numerical accuracy (MAE/MSE) drops significantly. This observation suggests that not all intrinsic features, particularly numerical range, can be well preserved through the attribution process.
>
> #### Table 3. Ablation study of $x\_{h,0}$ on Synth dataset. $\uparrow$ ($\downarrow$) means the higher (lower), the bettter.
>
> |Setting|$\downarrow$MAE|$\downarrow$MSE|$\uparrow$DTTC-I|$\uparrow$DTTC-E|
> |:--:|:--:|:--:|:--:|:--:|
> |TADiff|$0.54\_{\pm 0.00}$|$0.51\_{\pm 0.00}$|$15.37\_{\pm 0.03}$|$78.40\_{\pm 0.13}$|
> |TADiff w/o $x\_{h,0}$|$1.37\_{\pm 0.02}$|$2.47\_{\pm 0.05}$|$14.62\_{\pm 0.02}$|$78.44\_{\pm 0.11}$|
>
>
> > **Q3** How is the conditioning signal $c$ extracted from text? If a specific embedding model is used, please provide details for reproducibility. Also, how sensitive is overall performance to the quality of this embedding model?
>
> Thanks for your questions.
>
> We adopt the tokenizer of Long-CLIP [1] as our text tokenizer, and adopt a text encoder with 2 transformer layers. The text encoder is trained from scratch since the semantics in time series caption are compact and concise, a simple text encoder is enable to learn meaningful information. We have added the details of the text encoder we used in the latest version of the paper, the code will also be released once the paper is accepted.
>
> We also conduct an experiment to study the influence of the text embedding model. We compare our text encoder with Long-CLIP [1], BERT [2], and GPT-2 [3] on the Synth dataset. All pretrained text encoders are frozen and followed by a trainable transformer layer. The results in Table 4 show that our method is not sensitive to the choice of text embedding model. A simple 2-layer transformer is sufficient for modeling the text conditions, and this lightweight text encoder offers substantial advantages in both training and inference efficiency.
>
> #### Table 4. Influence of different text embedding models.
> |Text encoder|$\downarrow$MAE|$\downarrow$MSE|$\uparrow$DTTC-I|$\uparrow$DTTC-E|
> |:--:|:--:|:--:|:--:|:--:|
> |2-layer transformer|$0.54\_{\pm 0.00}$|$0.51\_{\pm 0.00}$|$15.37\_{\pm 0.03}$|$78.40\_{\pm 0.13}$|
> |Long-CLIP|$0.53\_{\pm 0.00}$|$0.50\_{\pm 0.00}$|$15.34\_{\pm 0.00}$|$78.32\_{\pm 0.06}$|
> |Bert|$0.54\_{\pm 0.01}$|$0.51\_{\pm 0.01}$|$15.33\_{\pm 0.03}$|$78.11\_{\pm 0.03}$|
> |GPT2|$0.54\_{\pm 0.00}$|$0.50\_{\pm 0.01}$|$15.30\_{\pm 0.03}$|$77.86\_{\pm 0.07}$|
>
> [1] Long-CLIP: Unlocking the Long-Text Capability of CLIP. ECCV 2024.
>
> [2] BERT: Pre-training of Deep Bidirectional Transformers for Language Understanding. ACL 2019.
>
> [3] Language Models are Unsupervised Multitask Learners. 2019.

---

> ### Author Response · Authors · 2025-11-21
> **Rebuttal by Authors (part 5 of 5)**
>
> > **Q4** "Finding 2" on page 8 cites the wrong table(s). Please correct the reference.
>
> Thanks for your questions.
>
> If we understand your point correctly, we believe the tables referenced in Finding 2 are accurate. The confusion may stem from a mismatch in interpretation between the reviewer and us, and we appreciate the opportunity to clarify this.
> - The Synth dataset includes the setting of forecasting under diverse future conditions. Since the generative process of the Synth dataset is fully controlled by us, every future condition has a corresponding ground-truth future time series. Therefore, MAE and MSE are available for evaluation on all data in the Synth dataset, and we report these results in Table 1 of our paper.
> - In contrast, the real-world datasets differ between the factual setting and the counterfactual setting, where the ground truth of the future time series is unavailable under counterfactual setting. Thus, only DTTC-I and DTTC-E are applicable for evaluation on the real-world datasets for counterfactual forecasting. For this reason, we present the forecasting results under the counterfactual setting in Table 2 of our paper.
>
> Finding 2 in our paper try to prove that our method (TADiff) is able to adapt to counterfactual conditions. Therefore, we take the results of synthetic data in Table 1 and real-world in Table 2 as the evidence.
>
> > **Q5:** This is more of a suggestion, the related work section should also cover causal inference fundamentals: what guarantees the field provides and how this work is situated among existing approaches.
>
> Thanks for your suggestions. We agree that it is important to clarify our position within the field of causal inference.
>
> We discuss several works on causal inference in Sec.2.2 “Counterfactual modeling”, including their applications in reasoning, image editing, and forecasting. We also include a causal inference method [4] as one of our baselines and compare its performance with ours.
>
> This work (TADiff) can be viewed as an application-oriented extension of causal inference within the domain of time series forecasting. The principles of causal inference have inspired our formulation of intrinsic features and external conditions in time series. We have added a statement in the related work section of the latest version of the paper to further clarify the connection between our work and the broader field of causal reasoning.
>
> [4] Causal transformer for estimating counterfactual outcomes. ICML 2022.

---

> > ### Comment · Reviewer_FZhN · 2025-11-25
> >
> > This message is to acknowledge that I have read the author's response carefully. Thank you for addressing my main concerns. I believe the added clarifications (especially on the synthetic data experiment) are very valuable. I believe the general idea of this paper is very valuable, the only aspect I am not fully convinced of is the benefit of the counterfactual fine-tuning. I think it has potential to be a critical strategy (for the proposed method and all other benchmarks), but the presented results are not convincing enough. Overall, I think the idea of the paper is valuable and the updated manuscript is improved.

---

> > > ### Author Response · Authors · 2025-11-26
> > > **About remaining question**
> > >
> > > Thank you for recognizing the contributions of our work and the improvements made during the rebuttal. In this response, we focus on addressing the remaining concern regarding the improvement of the counterfactual finetuning.
> > >
> > > Our evidences of counterfactual finetuning improvements are from two perspectives as detailed as below: (i) existing ablation study and significance analysis, (ii) further ablation study on an additional dataset.
> > >
> > > (i) **The existing ablation study has already demonstrated the effectiveness of counterfactual finetuning**. As discussed in Finding 3 of our paper and in our previous response to W4, we conducted paired t-tests to assess whether the performance differences observed in the ablation study are statistically significant. The results show that the improvements in semantic consistency (DTTC-I and DTTC-E) between the given counterfactual condition and forecasted time series are statistically significant, while numerical accuracy (MAE and MSE) remains uncompromised.
> > >
> > > (ii) **Further ablation on the Exchange dataset confirms the benefits of counterfactual finetuning**. To provide a more comprehensive verification, we conducted an additional ablation study on the Exchange dataset (detailed in Appendix B.3.1 of our paper) to evaluate the effect of counterfactual finetuning. As shown in Table 1 below, the results are consistent with the conclusions of Finding 3: counterfactual finetuning improves semantic consistency while preserving numerical accuracy.
> > >
> > > We hope the above analysis resolves the reviewer's concern regarding counterfactual finetuning, and we look forward to receiving your feedback or any further specific questions.
> > >
> > > #### Table 1. Ablation study of counterfactual finetuning on Exchange dataset. CF represents the counterfactual finetuning.
> > > |Setting|Factual|Factual|Factual|Factual|Counterfactual|Counterfactual|
> > > |:--:|:--:|:--:|:--:|:--:|:--:|:--:|
> > > |Method|$\downarrow$MAE|$\downarrow$MSE|$\uparrow$DTTC-I|$\uparrow$DTTC-E|$\uparrow$DTTC-I|$\uparrow$DTTC-E|
> > > |TADiff (ours)|$0.15_{\pm 0.01}$|$0.04_{\pm 0.00}$|$17.73_{\pm 0.04}$|$216.03_{\pm 3.11}$|$17.30_{\pm 0.04}$|$213.94_{\pm 2.18}$|
> > > |TADiff w/o CF|$0.15_{\pm 0.00}$|$0.06_{\pm 0.03}$|$16.64_{\pm 0.02}$|$210.27_{\pm 1.69}$|$16.50_{\pm 0.03}$|$209.07_{\pm 1.33}$|

---

### Official Review · Reviewer_iGqX · 2025-10-31

**Soundness:** 2
**Presentation:** 3
**Contribution:** 2
**Rating:** 4
**Confidence:** 3

**Summary:**

The paper introduces TADIFF, a diffusion-based model for time series forecasting conditioned on textual descriptions of future scenarios. It tackles the challenge of making accurate forecasts when future conditions differ from historical data (out-of-distribution). The authors also propose new evaluation metrics to assess forecast consistency. The work highlights the need for models and metrics that can handle complex, stochastic, and novel future conditions.

**Strengths:**

TADIFF is designed to generalize to out-of-distribution scenarios by conditioning forecasts on textual descriptions of future conditions, attempting to overcome the limitations of models that rely solely on historical data.

The paper introduces new evaluation metrics and a framework to assess forecast consistency with both historical patterns and future textual conditions, even in the absence of ground truth, addressing a key challenge in counterfactual forecasting.

**Weaknesses:**

- The proposed evaluation metrics are not fully explained or contrasted with traditional metrics like MAE/MSE. It is unclear what is gained by using these new metrics, and how they relate to or improve upon standard accuracy-based measures. More concrete examples and justification are needed.
- The process for constructing counterfactual datasets is not well detailed.
- In Table 3 and other results, it is unclear whether the observed differences are statistically significant. The absence of standard error metrics (like MAE/MSE) for counterfactuals makes it difficult to compare with existing baselines.
- The diffusion-based approach and the new evaluation framework add complexity. It is not clear if the benefits outweigh the additional implementation and computational costs for practitioners.

**Questions:**

- How do you ensure future conditions used for counterfactuals are realistic, meaningful, or representative of plausible scenarios.
- Can you expand on the proposed metrics? Maybe elaborate more with examples or more information about why accuracy is not as good as consistency.
- Can you provide more detials on Table 3 and if the difference is significant? Why no MAE or MSE for counterfactuals?

---

> ### Author Response · Authors · 2025-11-21
> **Rebuttal by Authors (part 1 of 3)**
>
> > **W1.1** The proposed evaluation metrics are not fully explained or contrasted with traditional metrics like MAE/MSE.
>
> Thanks for your questions.
>
> The proposed evaluation metrics DTTC-I and DTTC-E offer several advantages over MAE/MSE.
> - MAE and MSE require access to the ground-truth future time series, which are unavailable in the context of counterfactual forecasting.
> - In contrast, DTTC-I and DTTC-E enable evaluation even when ground-truth future sequences are absent, which directly addresses the core challenge of forecasting under counterfactual conditions.
>
> We introduce the limitations of MAE and MSE in Sec.1 to motivate the need for new evaluation metrics. The advantages of DTTC scores and their evaluation procedure are further detailed in Sec.3.4.
>
> > **W1.2** It is unclear what is gained by using these new metrics, and how they relate to or improve upon standard accuracy-based measures. More concrete examples and justification are needed.
>
> Thanks for your questions.
>
> Compared with numerical accuracy metrics (MAE/MSE), semantic consistency metrics (DTTC-I/DTTC-E) provide a more appropriate evaluation for forecasting under diverse counterfactual conditions, where inherent uncertainty makes point-wise numerical comparison less meaningful. We discuss these points in Sec.3.4.
>
> We have demonstrated that DTTC scores can reliably evaluate the quality of counterfactual forecasting. As shown in Finding 7, the DTTC model achieves high retrieval accuracy, indicating that it captures the semantic consistency between forecasts and historical patterns, as well as between forecasted time series and specified future conditions.
>
> We have also shown that the semantic consistency metrics (DTTC-I/DTTC-E) do not conflict with numerical accuracy metrics (MAE/MSE). As presented in Finding 3, using DTTC scores as optimization objectives improves the DTTC scores themselves while preserving MAE and MSE performance.
>
> All the discussions and results above (also included in the paper) collectively support that DTTC scores are reliable metrics for counterfactual forecasting and effectively address the dilemma arising from the absence of future ground truth.
>
> > **W2** The process for constructing counterfactual datasets is not well detailed.
>
> Thanks for your suggestions
>
> The details of constructing counterfactual data are provided in Sec.3.5, Sec.4.1, and Appendix B. We briefly summarize the process as below.
> * For the synthetic dataset, both factual and counterfactual data are generated by constructing diverse future conditions and corresponding future time series based on the same historical sequence, as discussed in Sec.4.1 and Appendix B.2.
> * For the real-world datasets, the counterfactual data construction procedure is described in Sec.3.5 and consists of two steps:
>     - (i) For each historical time series, randomly sample $M$ candidate future conditions from the factual dataset.
>     - (ii) Filter these candidates and select the future condition with the highest similarity to the historical condition.
>
> If you have any further specific questions, please feel free to let us know.
>
> > **W3.1** In Table 3 and other results, it is unclear whether the observed differences are statistically significant.
>
> Thanks for your questions.
>
> In Table 1–3 of our paper, we report the mean and standard deviation of results over 3 runs with different random seeds. The performance improvements exceed the corresponding standard deviations, indicating that the differences are statistically meaningful. We further conducted paired t-tests between our method (TADiff) and the baseline with the second-best performance. The resulting p-values are mostly below 0.05, providing additional evidence that the performance differences are statistically significant.
>
> > **W3.2** The absence of standard error metrics (like MAE/MSE) for counterfactuals makes it difficult to compare with existing baselines.
>
> As discussed in Sec.1, Sec.3.4, and the response to **W1**, standard error metrics (such as MAE/MSE) face a fundamental dilemma when evaluating forecasting under counterfactual settings, because the ground truth of future time series is unavailable. This motivates us to propose the new metrics DTTC-I and DTTC-E, which address this limitation and enable evaluation under counterfactual conditions.
>
> Given the demonstrated reliability of DTTC scores in Finding 7, we believe that comparing our method with existing baselines using DTTC scores is both fair and reasonable.

---

> ### Author Response · Authors · 2025-11-21
> **Rebuttal by Authors (part 2 of 3)**
>
> > **W4** The diffusion-based approach and the new evaluation framework add complexity. It is not clear if the benefits outweigh the additional implementation and computational costs for practitioners.
>
> Thanks for your questions.
>
> Our method (TADiff) achieves strong forecasting performance under both factual and counterfactual conditions. As shown in Finding 1 and 2 of our paper, TADiff demonstrates substantial improvements over all baselines.
>
> The efficiency of TADiff is also acceptable. As reported in Appendix E.2, the inference speed of our diffusion-based model is faster than existing diffusion-based and foundation-model baselines, which is also accepted in most real-world scenarios.
>
> TADiff has a low reproduction cost. As described in Sec.6 “Reproducibility Statement,” we provide detailed instructions for reproducing both the TADiff and DTTC models. We will also release our code and datasets once the paper is accepted.
>
> > **Q1** How do you ensure future conditions used for counterfactuals are realistic, meaningful, or representative of plausible scenarios.
>
> Thanks for your questions.
>
> We believe it is important to ensure and validate that the constructed counterfactual conditions remain plausible in real scenarios. To this end, we apply several operations to improve the quality of counterfactual data and conduct an experiment to assess its plausibility.
>
> (i) During counterfactual data construction, we aim to ensure that the future conditions differ from the historical patterns while avoiding significant conflicts between them. As detailed in Sec.3.5, the former is achieved by randomly sampling future conditions from factual data, and the latter is ensured through a similarity-based filtering step.
>
> (ii) We conduct an experiment to evaluate the quality of the counterfactual data.
> - We compare the similarity between historical and future conditions across factual data and constructed counterfactual data. Two construction strategies are analyzed: (a) random sampling combined with similarity filtering, which is the method we adopt, and (b) random sampling only.
> - The results are summarized in Table 1 below, where "factual" refers to the original factual data, "counterfactual" refers to the data constructed using our method, and "random" refers to data generated by random sampling alone. The results show that our constructed counterfactual data exhibit higher similarity to the factual data than the purely random samples. This demonstrates that the counterfactual data produced by our method are plausible and share a distribution closer to factual data.
>
> #### Table 1. Similarity analysis between factual and two types of counterfactual data.
>
> |Datasets|$\text{Sim}\_{\text{factual}}$|$\text{Sim}\_{\text{counterfactual}}$|$\text{Sim}\_{\text{random}}$|
> |:--:|:--:|:--:|:--:|
> |ETTm1|165.59|155.96|104.83|
> |Exchange|294.85|287.68|110.63|
> |Traffic|86.12|84.47|72.64|
> |Weather|55.57|57.42|46.05|
>
> > **Q2** Can you expand on the proposed metrics? Maybe elaborate more with examples or more information about why accuracy is not as good as consistency.
>
> Thanks for your questions.
>
> We believe that semantic consistency metrics (DTTC-I/DTTC-E) are more appropriate than numerical accuracy metrics (MSE/MAE) for the following reasons:
> (i) Numerical accuracy metrics cannot evaluate forecasting under counterfactual conditions due to the absence of future ground truth, whereas semantic consistency metrics directly address this limitation. We discuss this in Sec.1 and Sec.3.4.
> (ii) Overemphasizing numerical accuracy may cause the model to overfit to meaningless patterns [1]. For example, in many scenarios (Weather, Traffic, etc.), what truly matters is the trend direction of the forecasts.
> - Consider a ground-truth of forecasting whose trend first rises and then falls. Numerical accuracy might prefer a smooth straight-line forecast over a forecast that accurately captures the correct up-then-down trend but exhibits a larger numerical scale difference. In such cases, the latter prediction learns the correct future dynamics, whereas the former simply overfits to an unrealistic pattern in pursuit of lower numerical error.
>
> Nevertheless, we believe that these two types of metrics evaluate forecasting quality from different perspectives, and ideally both should be considered together. However, due to the inherent limitations of counterfactual forecasting, numerical accuracy metrics cannot be used for evaluation in this setting.
>
> [1] Patch-wise Structural Loss for Time Series Forecasting. ICML 2025.

---

> ### Author Response · Authors · 2025-11-21
> **Rebuttal by Authors (part 3 of 3)**
>
> > **Q3** Can you provide more detials on Table 3 and if the difference is significant? Why no MAE or MSE for counterfactuals?
>
> Thanks for your questions.
>
> As detailed in the response of **W3**, we report the mean and standard deviation of results in 3 runs with different random seeds in Table 3 and conduct the paired t-tests to prove the significance of the differences.
>
> MAE and MSE are unable to evaluate the forecasting under counterfactual settings since the ground truth of the future time series are absent. These motivate us to propose the new metrics DTTC-I and DTTC-E, as discussed in Sec.3.4.

---

### Official Review · Reviewer_Tfgp · 2025-10-31

**Soundness:** 1
**Presentation:** 2
**Contribution:** 3
**Rating:** 4
**Confidence:** 4

**Summary:**

### *Summary*

- The authors propose a diffusion-based model for forecasting time series with additional textual information. Stage 1 estimates and removes the contribution to the series coming from c\_h by running the inverse of the diffusion denoising. Stage 2 uses this “intrinsic feature” and runs a diffusion denoising process on it to get a forecast. The authors also propose DTTC, a learned metric which disentangles intrinsic and extrinsic influence on a given series to estimate the forecast’s consistency with both. The authors use LLMs to generate textual information based on window statistics, which they use to train the diffusion model and DTTC. They also use this methodology to generate multiple counterfactuals, which demonstrates the generalization ability of their model.

### *Contributions*

- attribution-forecast paradigm
- Text-Attributive time series Diffusion (TADIFF) model (or framework, or paradigm)
- Novel metric approach

**Strengths:**

### *Originality*

- Strong novelty component with the modelling approach, DTTC and generalization potential

### *Quality*

- Many interesting findings.
- Hyperparameter robustness analysis is good.

### *Clarity*

- Well-structured text, informative result visuals (e.g. figures 3 and 5\)

### *Significance*

- Works on a timely problem, i.e. forecasting with essential textual information

**Weaknesses:**

### *Originality*

- The problem setting of forecasting with relevant textual information about future events, such that the “future dynamics diverge substantially from past observations”, is not a novel problem setting (see [https://arxiv.org/abs/2410.18959](https://arxiv.org/abs/2410.18959) and [https://www.arxiv.org/abs/2508.09904](https://www.arxiv.org/abs/2508.09904), both of which treat text about future conditions). Augmenting data to create diverse sets of plausible futures for a single series is interesting from a comparative evaluation perspective, but is more akin to a data augmentation strategy, not a novel problem setting.
- Contemporaneous work that is highly related: [https://openreview.net/forum?id=Zbt44sC4tE](https://openreview.net/forum?id=Zbt44sC4tE) (I won't penalize for this)

### *Quality*

- It’s unclear how you make sure that the DTTC-I and DTTC-E metrics remain grounded and generalize (see Questions).
- Finding 1: these datasets are saturated [https://www.arxiv.org/abs/2510.02729](https://www.arxiv.org/abs/2510.02729). It’s unclear what SOTA on these datasets actually means.
- Insufficient detail to understand whether the results of finding 7 are sufficient to back up the claims of DTTC being a good metric.
- It’s unclear what the relevance of Leibniz’s Principle of Continuity has to do with general time series forecasting. Why would exchange rate time series be bound to “natural” changes (L199)?

### *Clarity*

- The framing of the contributions is inconsistent.
  - The intro frames the contributions as 1\) the task, 2\) the evaluation and 3\) the attribution paradigm.
  -  The conclusion frames the contributions as 1\) the task, 2\) the TADiff model and 3\) the evaluation metrics
  - The intro and abstract do not mention TADiff at all, despite this being a key contribution in the conclusion.
- The use of terms such as “framework”, “paradigm” and other vague, non-technical terms is confusing. I would avoid using these terms, as they make the assessment of the paper’s contributions more difficult. If it’s a model architecture \+ training strategy, please just say that.
- Figure text is too small, illegible at normal document size
- Figure 1 is good, but the caption is bare. There are textual elements in the figure that are not discussed in the caption, e.g. the background information on traffic volume. Is that input, or just a description for the example? The caption should clarify this, and ideally “walk through” the figure.
- The caption for figure 2 is uninformative.
- If DTTC is a major contribution to the paper, the training details should not be solely in the appendix. These details are too important to understand whether DTTC makes sense as an approach to be left in the appendix. In fact, this relates to a more general comment on criticism: there are too many findings in the main paper, which prevents you from analyzing any in depth. I would prioritize 2 or 3 main findings, discuss those in the main text, and relegate the rest to “Additional Results” in the appendix.
- Inconsistent terminology: why flip to “text-attribution” in section 4.3? This is inconvenient for ctrl-F

**Questions:**

### *change your opinion*

- The concepts are good, but the *clarity* of the article is significantly impeded.
  -  Key details on dataset generation that are relevant to generalization capacity are buried in the appendix,
  - non-technical terms are used to discuss model architectures,
  - there are too many results in the main text to discuss in depth,
  - the figures are not clear,
  - key training details are relegated to the appendix,
  - terminology is inconsistent.
- There is key missing related work that undermines the proposed contributions, which hinders the alignment between the paper’s claims and its contributions/results.
  - I would shift away from claiming that the problem setting is a contribution and cite the appropriate work instead (Context is Key, Time-MMD, etc.)
    - The paper should reframe the counterfactual generation as an LLM-based data augmentation strategy because that’s what it is: you use an LLM to generate text that accompanies the series, and show that your architecture works well when given access to this text.
    - The contributions should avoid overclaims on novel problem settings, and instead focus around the article’s actual contributions:
      - model
      - DTTC
      - data generation strategy
      - generalization guarantees

By reframing the article to align the claims with the contributions and citing appropriate relevant work, I can see myself raising the score to a 6\. By performing additional generalization experiments and seeing how well the approach generalizes, the score could raise to an 8, although I would expect significantly more generalization results, such as:

- Results on MM-TSFLib and/or Context is Key
- Assessment of overfitting to LLM-generated textual patterns by testing different LLM families for textual generation

### *clarify a confusion*

- I’m a bit confused about the *order* in which the different components are trained, and how you ensure that evaluation using DTTC is clean: you use DTTC to generate the counterfactual datasets, but then you estimate generalization ability based on DTTC scores in Table 2\. If you’re optimizing your finetuning generalization based on DTTC scores, then you can’t really use it to evaluate generalization ability, right? Could you fit your model \+ DTTC to a subset of datasets, and then test on some held-out datasets to see whether the metric and model generalization results still hold on unseen datasets/patterns?
- Where do you get the original textual condition c\_h? The datasets you use do not have accompanying text. Is this what is described in B.3.1? If so, I’d put that in the main text as it’s quite important for understanding the dataset generation and limitations of generalization ability. How do you avoid overfitting to an LLM’s generated text format? One way to do this would be to demonstrate performance on a separate benchmark, such as Context Is Key ([https://arxiv.org/abs/2410.18959](https://arxiv.org/abs/2410.18959)), which has already filtered for ensuring the context is highly relevant.

### *address a limitation*

- The maintainers of these forecasting datasets (ETT, Exchange, Weather, possibly Traffic) have publicly stated that they are saturated: [https://www.arxiv.org/abs/2510.02729](https://www.arxiv.org/abs/2510.02729). What is the value of building on top of these datasets?

---

> ### Author Response · Authors · 2025-11-21
> **Rebuttal by Authors (part 1 of 8)**
>
> Thank you for your detailed and valuable comments. To address all of the reviewer's questions thoroughly, our response is a bit lengthy. We apologize in advance and hope it does not impose too much burden on the reviewer.
>
> > **W1.1** The problem setting of forecasting with relevant textual information about future events, such that the “future dynamics diverge substantially from past observations”, is not a novel problem setting (see https://arxiv.org/abs/2410.18959 and https://www.arxiv.org/abs/2508.09904, both of which treat text about future conditions). Augmenting data to create diverse sets of plausible futures for a single series is interesting from a comparative evaluation perspective, but is more akin to a data augmentation strategy, not a novel problem setting.
>
> Thanks for your suggestions and questions.
>
> Although our proposed text-based counterfactual forecasting shares certain similarities with forecasting conditioned on future textual event information, there are also fundamental differences. Our newly proposed task is distinct from existing works in several key aspects.
> - Compared with forecasting based on future events, we do not assume a single deterministic future, which are the base assumption of the mentioned works. Instead, we consider diverse future conditions, covering both factual and counterfactual scenarios which are more common in real world.
> - Compared with traditional counterfactual modeling, our task incorporates textual conditions, which offer more granular and diverse control signals for generating future forecasts and have not been well studied in literature.
>
> Reducing our task to a simple form of data augmentation for future-condition forecasting would be inappropriate, as our setting introduces several unique challenges that prior work has not addressed:
> - Models trained solely on observed real-world data often struggle to adapt when placed under diverse counterfactual conditions.
> - The absence of ground-truth observations in counterfactual scenarios creates a fundamental barrier to evaluating forecasting quality.
>
> The novelty and contribution of our proposed task have been acknowledged by reviewers kMJW, iGqX, and FZhN. In addition, per your valuable comments, we have added the two mentioned works (https://arxiv.org/abs/2410.18959 and https://www.arxiv.org/abs/2508.09904) to our related works section. Clarifying the differences between their tasks and ours will help readers fully understand our contributions.
>
> > **W1.2** Contemporaneous work that is highly related: https://openreview.net/forum?id=Zbt44sC4tE (I won't penalize for this)
>
> Thanks for the related work you provided. We are glad to see that we are not the only ones focusing on counterfactual forecasting in time series, which further demonstrates that this is an important and valuable research direction.
>
> Although this work shares a similar task definition with ours, there remain substantial differences in both methodological contributions and evaluation frameworks.
> - “WHAT IF TSF” (the paper provided by Reviewer Tfgp) constructs a comprehensive benchmark for time series counterfactual forecasting. However, its evaluation of counterfactual forecasting is limited to directional accuracy, which cannot capture the rich semantics encoded in textual conditions. In contrast, our work employs novel DTTC scores to assess the semantic consistency between forecasts and text conditions, offering a more fine-grained and comprehensive evaluation.
> - “WHAT IF TSF” only evaluates the counterfactual forecasting capabilities of several existing LLMs. In our work, we further introduce the model TADiff, which is specifically designed to better address counterfactual forecasting under text conditions.
>
> Thanks again for sharing this paper. Although we cannot include it in the related work section at this stage, we will continue to follow and study this line of research.

---

> > ### Author Response · Authors · 2025-11-21
> > **Rebuttal by Authors (part 7 of 8)**
> >
> > > **Q4.2** How do you avoid overfitting to an LLM’s generated text format? One way to do this would be to demonstrate performance on a separate benchmark, such as Context Is Key (https://arxiv.org/abs/2410.18959), which has already filtered for ensuring the context is highly relevant.
> >
> > Thanks for your questions.
> >
> > There is no LLM involved in accompanying text generation, but there is still risk of overfitting to the text template. Therefore, we conducted an experiment to show that TADiff truly learns information from the meaningful content of the text rather than relying on the template-specific wording.
> > - Experiment setting. We manually classified the tokens appearing in the generated text into two categories: relevant and irrelevant. Relevant tokens contain meaningful semantics related to the time series, while irrelevant tokens mainly serve to maintain valid sentence structure. For example, in the sentence “The trend direction is up.”, the tokens “trend”, “direction”, and “up” are relevant, whereas “The”, “is”, and “.” are irrelevant. We independently masked relevant and irrelevant tokens at varying ratios and analyzed the resulting performance degradation. We conducted the experiment on the factual data of the ETTm1 and Exchange datasets, where both datasets use generated text.
> > - Experiment results. As shown in Table 1–4 below, masking irrelevant tokens leads to significantly smaller performance drops compared with masking relevant tokens. Notably, for MAE and MSE, masking 70% of the irrelevant tokens still yields performance comparable to masking only 10% of the relevant tokens. These results demonstrate that TADiff indeed learns from the meaningful semantic information in the generated text rather than overfitting to the constructed templates.
> > - We have also added visualization results in the latest version of the paper for clearer interpretation. Please refer to Fig.8 for more details.
> >
> > In addition to the datasets with generated text, our experiment also includes the Weather dataset, which is a real-world dataset with highly relevant text from expert annotation. This further demonstrates that our method performs well under real-world text conditions.
> >
> > #### Table 1. Masking relevant tokens on ETTm1 dataset. $\uparrow$ ($\downarrow$) means the higher (lower), the bettter.
> > |Relevant mask ratio|$\downarrow$MAE|$\downarrow$MSE|$\uparrow$DTTC-I|$\uparrow$DTTC-E|
> > |:--:|:--:|:--:|:--:|:--:|
> > |0.0|$0.57\_{\pm 0.04}$|$0.76\_{\pm 0.10}$|$10.01\_{\pm 0.03}$|$146.91\_{\pm 1.78}$|
> > |0.1|$0.73\_{\pm 0.03}$|$1.26\_{\pm 0.16}$|$10.03\_{\pm 0.02}$|$140.14\_{\pm 2.01}$|
> > |0.3|$0.95\_{\pm 0.01}$|$1.87\_{\pm 0.07}$|$9.66\_{\pm 0.03}$|$130.44\_{\pm 1.42}$|
> > |0.5|$1.15\_{\pm 0.01}$|$2.54\_{\pm 0.01}$|$9.22\_{\pm 0.08}$|$121.61\_{\pm 0.40}$|
> > |0.7|$1.31\_{\pm 0.03}$|$3.07\_{\pm 0.09}$|$8.93\_{\pm 0.19}$|$114.77\_{\pm 1.01}$|
> >
> > #### Table 2. Masking irrelevant tokens on ETTm1 dataset.
> > |Irrelevant mask ratio|$\downarrow$MAE|$\downarrow$MSE|$\uparrow$DTTC-I|$\uparrow$DTTC-E|
> > |:--:|:--:|:--:|:--:|:--:|
> > |0.0|$0.57\_{\pm 0.04}$|$0.76\_{\pm 0.10}$|$10.01\_{\pm 0.03}$|$146.91\_{\pm 1.78}$|
> > |0.1|$0.62\_{\pm 0.04}$|$0.81\_{\pm 0.19}$|$10.01\_{\pm 0.00}$|$145.07\_{\pm 2.17}$|
> > |0.3|$0.70\_{\pm 0.00}$|$1.08\_{\pm 0.06}$|$9.93\_{\pm 0.03}$|$140.75\_{\pm 2.79}$|
> > |0.5|$0.78\_{\pm 0.05}$|$1.28\_{\pm 0.10}$|$9.88\_{\pm 0.07}$|$131.03\_{\pm 4.89}$|
> > |0.7|$0.82\_{\pm 0.08}$|$1.35\_{\pm 0.20}$|$9.77\_{\pm 0.05}$|$125.54\_{\pm 7.03}$|
> >
> > #### Table 3. Masking relevant tokens on Exchange dataset.
> > |Relevant mask ratio|$\downarrow$MAE|$\downarrow$MSE|$\uparrow$DTTC-I|$\uparrow$DTTC-E|
> > |:--:|:--:|:--:|:--:|:--:|
> > |0.0|$0.15\_{\pm 0.01}$|$0.04\_{\pm 0.00}$|$17.37\_{\pm 0.04}$|$216.03\_{\pm 3.11}$|
> > |0.1|$0.18\_{\pm 0.00}$|$0.07\_{\pm 0.01}$|$17.09\_{\pm 0.07}$|$192.78\_{\pm 0.35}$|
> > |0.3|$0.23\_{\pm 0.01}$|$0.17\_{\pm 0.05}$|$16.84\_{\pm 0.07}$|$166.39\_{\pm 2.88}$|
> > |0.5|$0.33\_{\pm 0.03}$|$0.45\_{\pm 0.19}$|$16.49\_{\pm 0.12}$|$139.75\_{\pm 4.90}$|
> > |0.7|$0.40\_{\pm 0.05}$|$0.92\_{\pm 0.55}$|$16.05\_{\pm 0.28}$|$122.30\_{\pm 8.17}$|
> >
> > #### Table 4. Masking irrelevant tokens on Exchange dataset.
> > |Irrelevant mask ratio|$\downarrow$MAE|$\downarrow$MSE|$\uparrow$DTTC-I|$\uparrow$DTTC-E|
> > |:--:|:--:|:--:|:--:|:--:|
> > |0.0|$0.15\_{\pm 0.01}$|$0.04\_{\pm 0.00}$|$17.37\_{\pm 0.04}$|$216.03\_{\pm 3.11}$|
> > |0.1|$0.15\_{\pm 0.00}$|$0.05\_{\pm 0.00}$|$17.23\_{\pm 0.01}$|$214.82\_{\pm 0.29}$|
> > |0.3|$0.17\_{\pm 0.00}$|$0.05\_{\pm 0.00}$|$17.15\_{\pm 0.05}$|$199.88\_{\pm 1.52}$|
> > |0.5|$0.19\_{\pm 0.00}$|$0.06\_{\pm 0.00}$|$17.08\_{\pm 0.05}$|$178.61\_{\pm 2.79}$|
> > |0.7|$0.19\_{\pm 0.00}$|$0.07\_{\pm 0.00}$|$16.98\_{\pm 0.03}$|$158.59\_{\pm 3.52}$|

---

> ### Author Response · Authors · 2025-11-21
> **Rebuttal by Authors (part 2 of 8)**
>
> > **W2.1** It’s unclear how you make sure that the DTTC-I and DTTC-E metrics remain grounded and generalize (see Questions).
>
> Thanks for your questions.
>
> It's necessary to demonstrate that the evaluation metric is both grounded and capable of generalization.
> (i) There are several details about the DTTC model that need to be clearly aligned between the reviewer and us. The DTTC model is dataset-specific, meaning that it is trained on the training split of each dataset and evaluated on the corresponding test split. Only factual data with paired time series and text are used during training.
> (ii) The experiment in Finding 7 is conducted on the test set of each dataset, which provides evidence that DTTC scores are grounded and generalized.
> - Grounded. Given the future time series, the DTTC model achieves high accuracy when retrieving the corresponding historical sequence and future text. This shows that DTTC scores effectively capture the consistency of forecasts under real-world scenarios.
> - Generalized. Since the DTTC model is trained on the training split and evaluated on the test split of each dataset, the results demonstrate that DTTC possesses generalization ability rather than overfitting to the training data.
>
> > **W2.2** Finding 1: these datasets are saturated https://www.arxiv.org/abs/2510.02729. It’s unclear what SOTA on these datasets actually means.
>
> Thanks for your question.
>
> “Accuracy Law” (the paper provided by the reviewer) presents an interesting observation regarding the saturation of forecasting performance on certain datasets. However, its conclusions cannot be directly transferred to our work, as the experimental settings differ in several important aspects.
> - “Accuracy Law” focuses on *unimodal time series forecasting*, whereas our work investigates text-based multimodal forecasting. The findings of the “Accuracy Law” further highlight the necessity of incorporating multimodal information in time series forecasting, which is fully aligned with the motivation of our paper.
> - “Accuracy Law” restricts their settings in forecasting on *certain conditions*, i.e., the history time series. While our work additionally considers forecasting under uncertain future conditions, which introduces challenges beyond those in the traditional forecasting task as that in the mentioned work. Moreover, our evaluation incorporates semantic consistency metrics (DTTC-I/DTTC-E), offering deeper insights than purely numerical accuracy measures (MAE/MSE).
>
> In summary, we believe it is premature to conclude that these datasets are saturated *in the context of multimodal counterfactual forecasting*, where the problem setting and evaluation dimensions are fundamentally different.
>
> > **W2.3** Insufficient detail to understand whether the results of finding 7 are sufficient to back up the claims of DTTC being a good metric.
>
> Thanks for your suggestions.
>
> As mentioned in the response to **W2.1**, Finding 7 demonstrates that DTTC scores are grounded and generalized metrics. The detailed experimental setup is as follows:
> - The DTTC model is dataset-specific. It is trained on the training split of each dataset independently and evaluated on the corresponding test split. This setup validates its generalization ability.
> - For each future time series $x\_f^{(1)}$, we use the DTTC model to retrieve the most similar historical sequence $x\_h^{(1)}$ from three candidates $\{x\_h^{(1)}, x\_h^{(2)}, x\_h^{(3)}\}$ and the most similar text condition $c\_f^{(1)}$ from three candidates $\{c\_f^{(1)}, c\_f^{(2)}, c\_f^{(3)}\}$. The reported retrieval accuracy demonstrates that DTTC successfully aligns the intrinsic features between historical and future series, as well as the semantics between future series and external conditions.
>
> We have provided additional details on these findings in the latest version of our paper.

---

> ### Author Response · Authors · 2025-11-21
> **Rebuttal by Authors (part 3 of 8)**
>
> > **W2.4** It’s unclear what the relevance of Leibniz’s Principle of Continuity has to do with general time series forecasting. Why would exchange rate time series be bound to “natural” changes (L199)?
>
> Thanks for your questions.
>
> We believe that time series forecasting shares certain conceptual principles with Leibniz’s Principle of Continuity:
> - In time series forecasting, it is commonly assumed [1] that future changes are, to some extent, a continuation of past changes. Historical sequences contain inherent structures or patterns, such as trends, cycles, seasonality, and fluctuation behaviors, which do not vanish abruptly due to external or accidental conditions.
> - Leibniz’s Principle of Continuity, as expressed in philosophy and mathematics, states that “there are no leaps in nature.” In other words, natural changes evolve smoothly and gradually, rather than in abrupt, discontinuous jumps.
>
> We refer to Leibniz’s Principle of Continuity to highlight the importance of intrinsic features in time series forecasting. These invariant intrinsic features are one of the reasons why many natural processes evolve smoothly under stable conditions.
>
> Such invariant intrinsic features also exist in exchange rates, as their evolution has been shown to exhibit mean-reversion behavior [2]. Although exchange rates are indeed sensitive to external conditions, they tend to revert toward a long-term average rather than deviate indefinitely. Therefore, ignoring their intrinsic features would be undesirable.
>
> [1] Forecasting: Principles and Practice. 2018.
>
> [2] Nonlinear Mean-Reversion in Real Exchange Rates: Toward a Solution To the Purchasing Power Parity Puzzles. International economic review 2001.
>
> > **W3.1** The framing of the contributions is inconsistent.
> (i) The intro frames the contributions as 1) the task, 2) the evaluation and 3) the attribution paradigm.
> (ii) The conclusion frames the contributions as 1) the task, 2) the TADiff model and 3) the evaluation metrics
> (iii) The intro and abstract do not mention TADiff at all, despite this being a key contribution in the conclusion.
>
> We apologize for the confusion caused.
>
> In fact, our contributions are consistent throughout the paper. The reviewers’ questions may stem from the following reasons:
> - Confusion between TADiff and the attribution-forecast paradigm.
>   - TADiff is the name of our model, which is a multimodal diffusion model built upon the attribution-forecast paradigm. We explained the origin of the name “TADiff” in the introduction.
>   - To avoid causing unnecessary difficulty for readers, we describe our contributions in introduction using the more intuitive term “attribution-forecast paradigm” rather than the abbreviation “TADiff” when referring to the model.
> - The ordering of contributions differs between sections.
>   - In the introduction, we first present the task definition and evaluation method so that readers can quickly grasp the newly proposed task and its associated challenges of evaluation.
>   - In the coculsion, once readers have a complete understanding of the work, we reorder the contributions into the more conventional sequence of task, method, and empirical evaluation.
>
> > **W3.2** The use of terms such as “framework”, “paradigm” and other vague, non-technical terms is confusing. I would avoid using these terms, as they make the assessment of the paper’s contributions more difficult. If it’s a model architecture + training strategy, please just say that.
>
> Thanks for your suggestions.
>
> We apologize for the confusion caused by the term “framework” and “paradigm”. To provide a more concise and intuitive understanding, we refine the description of method's contributions:
> - An attribution-forecasting multimodal diffusion model.
> - A training strategy that utilizes constructed counterfactual data for finetuning.
>
> We have update the usage of terms in the latest version of our paper.
>
> > **W3.3** Figure text is too small, illegible at normal document size
>
> Thanks for your suggestions.
>
> We have adjusted the text size in figure for better readability.
>
> > **W3.4** Figure 1 is good, but the caption is bare. There are textual elements in the figure that are not discussed in the caption, e.g. the background information on traffic volume. Is that input, or just a description for the example? The caption should clarify this, and ideally “walk through” the figure.
>
> Thanks for your suggestions.
>
> The textual elements in Figure 1 are divided into two categories: intrinsic features and external conditions, as indicated in the legend. The external conditions serve as inputs, while the intrinsic features are presented as domain knowledge.
>
> We have updated the caption of Figure 1 to provide a more detailed explanation of these textual elements.

---

> ### Author Response · Authors · 2025-11-21
> **Rebuttal by Authors (part 4 of 8)**
>
> > **W3.5** The caption for figure 2 is uninformative.
>
> Thanks for your suggestions.
>
> We have updated the caption of Figure 2, providing more detailed explanation of the elements in the figure.
>
> > **W3.6** If DTTC is a major contribution to the paper, the training details should not be solely in the appendix. These details are too important to understand whether DTTC makes sense as an approach to be left in the appendix. In fact, this relates to a more general comment on criticism: there are too many findings in the main paper, which prevents you from analyzing any in depth. I would prioritize 2 or 3 main findings, discuss those in the main text, and relegate the rest to “Additional Results” in the appendix.
>
> Thanks for your suggestions.
>
> We place the training details of DTTC model in appendix since:
> - Compared with the main contribution of our method TADiff, the DTTC model serves primarily as an evaluation tool, and its contribution is relatively minor. Nevertheless, we provide detailed explanations in Sec3.4 on how DTTC is used for evaluation, which is more helpful for readers to fully understand our work than the full training details of DTTC.
>
> Regarding the too many findings in the main text:
> - We have moved several less critical analyses to the appendix, such as model efficiency analysis and case studies.
> - We believe all extended analyses in main text are important for understanding how TADiff works. Given the extended length limit, we provide more comprehensive and detailed discussions.
>
> Per your suggestions, we have updated DTTC training and findings in the latest version of our paper.
>
> > **W3.7** Inconsistent terminology: why flip to “text-attribution” in section 4.3? This is inconvenient for ctrl-F.
>
> Thanks for your suggestions.
>
> We propose a new model, TADiff, whose full name is “Text-Attributive Time Series Diffusion.” The term “Text-Attribution” is simply the nominal form of “Text-Attributive.”
>
> Nevertheless, we have updated the usage of terminology to improve clarity and facilitate more effective keyword search.
>
> > **Q1** The concepts are good, but the clarity of the article is significantly impeded.
> (1) Key details on dataset generation that are relevant to generalization capacity are buried in the appendix,
> (2) non-technical terms are used to discuss model architectures,
> (3) there are too many results in the main text to discuss in depth,
> (4) the figures are not clear,
> (5) key training details are relegated to the appendix,
> (6) terminology is inconsistent.
>
> Thanks for your suggestions.
>
> We have updated the writing of the paper in the latest version according to the reviewer’s advice. The revisions can be summarized into two categories:
> - We have allocated appropriate and sufficient space in the main text for dataset construction, DTTC training, and the analysis of key findings.
> - We have refined the terminology usage and improved the clarity and details of the figures.
>
> > **Q2.1** There is key missing related work that undermines the proposed contributions, which hinders the alignment between the paper’s claims and its contributions/results. I would shift away from claiming that the problem setting is a contribution and cite the appropriate work instead (Context is Key, Time-MMD, etc.)
>
> Thanks for your questions.
>
> The task we propose is fundamentally different from tasks that forecast based on deterministic future conditions. Such tasks fail to capture the inherent uncertainty of future conditions and therefore struggle to adapt across diverse possible futures. This distinction has been thoroughly discussed in Sec 2.1 of our paper. Our proposed task introduces unique challenges in both modeling and evaluation, which we further elaborate on in the response to **W1.1**.
>
> Time-MMD has already been discussed in the related works and included as a baseline in our experiments. Context is Key is not included because it is a benchmark rather than a newly proposed method. However, we have added a discussion in the latest version regarding the differences in task definitions between Context is Key and our work.

---

> ### Author Response · Authors · 2025-11-21
> **Rebuttal by Authors (part 5 of 8)**
>
> > **Q2.2** (1) The paper should reframe the counterfactual generation as an LLM-based data augmentation strategy because that’s what it is: you use an LLM to generate text that accompanies the series, and show that your architecture works well when given access to this text.
>
> We believe the reviewer may have misunderstood our data construction process, and some clarification is necessary.
>
> There is no usage of LLMs in our data construction.
> - The unimodal real-world datasets are captioned using text templates and an external tool (tsfresh), which is a library for extracting time series features. These details are provided in Appendix B.
> - The counterfactual data is generated through random sampling from factual data, with the full procedure described in Sec. 3.5.
>
> We included the statement “The Use of Large Language Models” in Appendix A to clarify that LLMs are only used to extract structured attributes for the inputs of certain baselines. Therefore, summarizing our work as “an LLM-based data augmentation strategy” is inappropriate.
>
> > **Q2.3** (2) The contributions should avoid overclaims on novel problem settings, and instead focus around the article’s actual contributions: model, DTTC, data generation strategy, generalization guarantees.
>
> Summarizing our responses above, we believe our contributions are appropriately scoped and aligned with the actual advances presented in the paper. Specifically, they include:
> - The newly proposed task: We define the task of counterfactual time series forecasting under text conditions. This task differs from existing time series forecasting on future events, as it explicitly considers the uncertainty of future conditions. This introduces unique challenges in both modeling and evaluation.
> - The model and training strategy: We develop an attribution-forecasting multimodal diffusion model for counterfactual forecasting, and further enhance its adaptability using constructed counterfactual data.
> - The evaluation: We propose new semantic-consistency metrics to evaluate forecasting under counterfactual conditions. These metrics address the dilemma of assessing counterfactual forecasts in the absence of ground-truth future observations.
>
> We will further clarify these contributions and refine the presentation in the revised version of the paper to avoid any confusion on these aspects.
>
> > **Q3** I’m a bit confused about the order in which the different components are trained, and how you ensure that evaluation using DTTC is clean: you use DTTC to generate the counterfactual datasets, but then you estimate generalization ability based on DTTC scores in Table 2. If you’re optimizing your finetuning generalization based on DTTC scores, then you can’t really use it to evaluate generalization ability, right? Could you fit your model + DTTC to a subset of datasets, and then test on some held-out datasets to see whether the metric and model generalization results still hold on unseen datasets/patterns?
>
> Thanks for your questions.
>
> There are several points that need to be clarified:
> - **DTTC and TADiff are dataset-specific.** Both factual and counterfactual data are split into train, validation, and test sets. The DTTC model is trained on the train set of factual data. TADiff is trained on the train set of factual data and finetuned on the train set of counterfactual data. Evaluation is performed on the test sets of both factual and counterfactual data.
> - **Definition of generalization.** The train and test splits contain different combinations of historical sequences and future conditions. We define generalization as training on the "train split" and evaluating on the "test split", rather than cross-domain generalization.
> - **Optimizing and evaluating with the same metrics is valid.** We have shown in Finding 7 that DTTC scores are reliable metrics for evaluating the quality of counterfactual forecasting. In traditional forecasting, MAE and MSE serve both as optimization objectives and evaluation metrics. Finding 3 further demonstrates that DTTC scores are not in conflict with MAE/MSE, as finetuning on DTTC scores does not degrade performance on MAE/MSE.
>
> We have refined the definition of generalization, clarified the description of the data splits, and specified which sets the models are trained and evaluated on in the latest version of our paper.

---

> ### Author Response · Authors · 2025-11-21
> **Rebuttal by Authors (part 6 of 8)**
>
> > **Q4.1** Where do you get the original textual condition $c\_h$? The datasets you use do not have accompanying text. Is this what is described in B.3.1? If so, I’d put that in the main text as it’s quite important for understanding the dataset generation and limitations of generalization ability.
>
> Thanks for your questions.
>
> For the unimodal real-world datasets consisting of only time series data, we generate the accompanying text using an external tool (tsfresh) and text templates, with no LLM involved. The details are provided in Appendix B.3.1. This method of text generation is inspired by [3], but we did not discuss it in the main text as it is not the primary contribution of our work.
>
> [3] VerbalTS: Generating Time Series from Texts. ICML 2025.

---

> ### Author Response · Authors · 2025-11-21
> **Rebuttal by Authors (part 8 of 8)**
>
> > **Q5** The maintainers of these forecasting datasets (ETT, Exchange, Weather, possibly Traffic) have publicly stated that they are saturated: https://www.arxiv.org/abs/2510.02729. What is the value of building on top of these datasets?
>
> Thanks for your question.
>
> While this article offers an interesting perspective on the potential saturation of forecasting performance on certain datasets, its conclusions cannot be directly applied to our study due to substantial differences in the experimental setup.
> - The work referenced by the reviewer focuses solely on unimodal time series forecasting, whereas our research centers on text-guided multimodal forecasting. In fact, their observations further underscore the importance of incorporating multimodal information in time series forecasting, which aligns closely with the motivation behind our work.
> - Furthermore, our method explicitly addresses forecasting under uncertain future conditions, a setting that poses challenges not present in traditional forecasting tasks. Our evaluation also goes beyond standard numerical metrics (MAE/MSE) by incorporating semantic consistency measures (DTTC-I/DTTC-E), providing a richer assessment of forecast quality.
>
> In conclusion, we consider it premature to assert that these datasets are saturated in the context of multimodal counterfactual forecasting, as the task formulation and evaluation metrics differ fundamentally from those examined in the referenced work.
>
> > **Overall Claim** By reframing the article to align the claims with the contributions and citing appropriate relevant work, I can see myself raising the score to a 6. By performing additional generalization experiments and seeing how well the approach generalizes, the score could raise to an 8, although I would expect significantly more generalization results.
>
> Thank you for the specific suggestions.
>
> We have included a discussion of the relevant papers in the updated related work section and clarified the task differences between our work and theirs. We also refined the statements of our contributions for better clarity in the latest version of the paper, as discussed in the responses to **W1.1,W3.1,Q2**.
>
> We added a sensitivity study to demonstrate that our method (TADiff) does not overfit to the text templates and can accurately capture the meaningful semantics contained in the text, as discussed in the response to **Q4**. In addition, we further detailed the experimental settings regarding dataset splitting, model training, and evaluation, as discussed in the response to **Q3**.
> It is worth emphasizing again that the generalization we consider is in-domain, where the test split contains unseen combinations of historical sequences and future conditions. Although we do not focus on cross-domain generalization, our experiments cover datasets from diverse sources, including a fully synthetic dataset, unimodal real-world datasets with generated text, and a multimodal real-world dataset with expert annotations. We believe these results support that TADiff is capable of handling counterfactual forecasting under complex conditions.

---

> > ### Comment · Reviewer_Tfgp · 2025-11-22
> > **Thank you for your thorough reply.**
> >
> > I would like to thank the authors for their exhaustive reply. The majority of my concerns have been addressed, so I'm raising my score to a 6. Out of concerns for brevity, I won't discuss each one individually, but the two main outstanding issues I see are 1) the problem setting, and 2) the textual generalization
> >
> > ## I still fail to see how the problem setting is novel.
> >
> > > Compared with forecasting based on future events, we do not assume a single deterministic future, which are the base assumption of the mentioned works. Instead, we consider diverse future conditions, covering both factual and counterfactual scenarios which are more common in real world.
> >
> > Any of the tasks with multiple counterfactuals can simply be decomposed into a set of forecasting tasks with a deterministic future and accompanying text. Therefore, the problem that a model is faced with (inputs are ts and text, output is ts that depends on ts and text) is not new: what changes is the composition of the dataset that the model is expected to forecast well over, which is why I described it as more of a data augmentation strategy. However, the problem the model faces is that it is expected to generate a time series forecast conditional on time series and text.
> >
> > > Compared with traditional counterfactual modeling, our task incorporates textual conditions, which offer more granular and diverse control signals for generating future forecasts and have not been well studied in literature.
> >
> > I agree with this in principle, but since you use templates, it becomes quite difficult to actually back the claim that your model can leverage these diverse symbols. Based on the explanation of tsfresh (see below) and the ablations you ran, I wouldn't expect the model to do well on tasks with subtle, diverse natural language in-the-wild, so it's hard to back the claim that this is what you are actually studying. Instead, the templated approach acts more like a "codebook" that the model is meant to interpret. Which leads me to the next point:
> >
> > ## Does this model generalize to non-templated text?
> >
> > Thank you for running these experiments, and my apologies for my misunderstanding of how the text is generated, I understand now that you used tsfresh with templates. Are you not concerned about your model overfitting to the textual templates? This seems like a big deal if you're using templates, your model won't generalize to other forms of natural language.
> >
> > The masking experiment shows that your model attends to the correct words. What it does *not* show is that this model can actually leverage natural language beyond the restricted set of templates you used: your model is essentially learning a "rote memorization" strategy where it ignores things like syntax and grammar, so I would expect it to have difficulty with a case like “The trend direction is up.” vs “The trend direction is not up." (unless this type of case was part of the template set), and then an even harder case with something like "sustained momentum from an optimistic outlook", or any other of a virtually limitless set of possible ways to describe (trend, up) in natural language.
> >
> > In my mind, this paper stands at a 6 right now:
> > - I'd like to better understand why you still think this problem setting is novel.
> > - The masking experiment shows where the attention lies, which is expected because you're using a template from which only a few terms are useful for disambiguating alternatives, but it doesn't show natural language understanding. Can you get an LLM or human to paraphrase the textual cues, and see if the model still performs well?
> >
> > Resolution of both of these would raise my score to an 8.

---

> > > ### Author Response · Authors · 2025-11-26
> > > **About remaining questions (part 2 of 2)**
> > >
> > > > Q2: Textual generalization: Does this model generalize to non-templated text?
> > >
> > > Yes, our model can generalize to non-templated text. We demonstrate this from two perspectives: (i) existing experiments on real-world datasets with natural, non-templated text, and (ii) additional experiments on datasets with LLM-refined text.
> > >
> > > (i) **Existing experiments have already demonstrated that TADiff is capable of handling non-templated text**. In particular, our evaluation includes the real-world Weather dataset, where the textual descriptions are annotated by human experts and align closely with what the reviewer refers to as "natural language". The text in the Weather dataset contains descriptions such as "the weather is clear initially, becoming cloudier as the hours pass", which provide rich and complex semantics that are difficult to capture using structured metadata. The experiments on the Weather dataset demonstrate that TADiff is capable of handling non-templated text.
> > >
> > > (ii) **An additional experiment on a dataset with LLM-refined text further demonstrates the model's ability to generalize to non-templated text**. Per the reviewer’s suggestion, we additionally used GPT-4.1 model to refine the templated-style texts in the Traffic dataset, and we provide an example below to illustrate the differences before and after refinement.
> > >
> > > * Templated-style text:
> > > ```
> > > The distribution of the value in time series is symmetrical, and has low kurtosis.
> > > For the overall shape, the time series decreases by time.
> > > The main season cycles is around 2 pi.
> > > At the end: the time series has a sharp downward trend.
> > > At the middle: the time series going up rapidly.
> > > At the beginning: there are 1 peaks, the time series has a slow upward trend.
> > > ```
> > > * LLM-refined text:
> > > ```
> > > The distribution of values over time is symmetric and exhibits low kurtosis. Overall, the series shows a decreasing trend. The primary seasonal cycle corresponds to a period of approximately 2π. At the beginning, the series features one peak with a gradual upward trend. In the middle segment, the series rises sharply. Toward the end, the series undergoes a pronounced downward trend.
> > > ```
> > > We trained the TADiff model and reconstructed the entire evaluation pipeline including the evaluation model DTTC on this LLM-refined dataset. We compared our method (TADiff) with several strong baselines (including the unimodal model Sundial [5] and the multimodal model TimeMMD [6]). As shown in Tables 2-3 below, the results indicate that TADiff still outperforms the baselines on both factual and counterfactual forecasting, further demonstrating that **our method generalizes well to non-templated text**.
> > >
> > > We hope the above experiments and analyses address the reviewer's concerns regarding generalization to non-templated text, we will revise our paper accordingly.
> > > And thank you again for your thoughtful comments and suggestions.
> > > We look forward to receiving your feedback or any further questions.
> > >
> > > #### Table 2. Factual forecasting on Traffic dataset.
> > > |Method|$\downarrow$MAE|$\downarrow$MSE|$\uparrow$DTTC-I|$\uparrow$DTTC-E|
> > > |:--:|:--:|:--:|:--:|:--:|
> > > |TADiff (ours)|$0.31_{\pm 0.00}$|$0.23_{\pm 0.00}$|$22.90_{\pm 0.05}$|$66.30_{\pm 0.15}$|
> > > |Sundial|$0.40_{\pm 0.00}$|$0.34_{\pm 0.00}$|$21.97_{\pm 0.00}$|$50.70_{\pm 0.04}$|
> > > |TimeMMD|$0.43_{\pm 0.01}$|$0.29_{\pm 0.01}$|$18.81_{\pm 0.21}$|$56.41_{\pm 0.43}$|
> > >
> > > #### Table 3. Counterfactual forecasting on Traffic dataset.
> > > |Method|$\uparrow$DTTC-I|$\uparrow$DTTC-E|
> > > |:--:|:--:|:--:|
> > > |TADiff (ours)|$22.87_{\pm 0.07}$|$64.20_{\pm 0.05}$|
> > > |Sundial|$21.94_{\pm 0.01}$|$50.63_{\pm 0.01}$|
> > > |TimeMMD|$18.71_{\pm 0.32}$|$53.46_{\pm 0.57}$|
> > >
> > > [5] Sundial: A Family of Highly Capable Time Series Foundation Models. ICML 2025.
> > >
> > > [6] Time‑MMD: Multi‑Domain Multimodal Dataset for Time Series Analysis. 2024.

---

> ### Author Response · Authors · 2025-11-26
> **About remaining questions (part 1 of 2)**
>
> We thank the reviewer for their recognition and positive feedback on our paper and rebuttal, as well as for raising the score to 6. We will continue addressing the remaining questions regarding the problem setting and the generalization ability of our model to textual inputs.
>
> > Q1: Problem setting: I still fail to see how the problem setting is novel.
>
> We demonstrate the novelty of our problem setting from four perspectives: (i) task definition, (ii) real-world motivation and examples, (iii) statistical analysis of the datasets used, and (iv) comparison with existing tasks.
>
> (i) The novelty of our proposed task lies not only in the diversity of future conditions but, more importantly, in the incorporation of **counterfactual setting**. **A counterfactual is defined as unseen combinations of historical sequences and future conditions that have not occurred in the real world**.
>
> (ii) **An example** might help clarify the differences from existing tasks and illustrate its **practical applications**.
> - Example: Consider a temperature time series from a drought-stricken region. In reality, due to prolonged drought, the region's temperature would remain high. Now, consider a counterfactual scenario where we disrupt the region's climate with artificial rainfall. An interesting question arises: what changes would occur in the region's temperature?
> - Explanation: In this scenario, the historical high-temperature pattern and the impact of artificial rainfall may have occurred independently in the real world, but this exact combination in this specific region has never actually happened. Therefore, no matter how many real-world data we sample, we cannot obtain such counterfactual data. This **aligns closely with real-world needs**, as humans often wish to predict the potential consequences of different decisions before those decisions are implemented.
>
> (iii) **The dataset we used matches the definition of counterfactual forecasting**. We believe that counterfactual settings can be quantified by how many combinations of conditions in the test split of counterfactual data appear in training split of factual data, which is the less the better. Specifically, we count the number of unique combinations of historical sequences and future conditions across different data splits on real-world Weather dataset. The results in Table 1 below show that only 80 attribute combinations appear in both the factual training split and the counterfactual testing split. This shared number (80) is dramatically smaller than the number of combinations in the factual training split (6,631) and the counterfactual testing split (4,066), indicating that most counterfactual scenarios consist of unseen combinations. This further validates both the necessity and the novelty of our problem setting.
>
> (iv) Our problem setting **faces unique challenges compared with existing tasks, which highlight the novelty of our problem setting**.
> - Compared with *forecasting based on future events* [1,2], our task additionally considers the counterfactual setting, which involves combinations of historical sequences and future conditions that have never occurred in the real world.
> - Compared with *forecasting on out-of-distribution (OOD) data* [3], counterfactual data cannot be obtained simply by expanding the data distribution. Because counterfactual scenarios never occur in reality, increasing the scale of real-world sampling cannot cover these cases.
> - Compared with *traditional counterfactual modeling* [4], our task incorporates textual conditions, which provide more granular and diverse control signals for generating future forecasts, and have not been well explored in the literature. We will further demonstrate our model’s ability to handle non-templated text in the response to the next question.
>
> We hope the above explanation addresses the reviewer's concerns regarding the novelty of our problem setting, and we look forward to receiving your feedback or any further questions.
>
> #### Table 1. The number of combination of historical sequences and future conditions on different splits of Weather dataset. $\text{F}$ represents the factual data, $\text{CF}$ represents the constructed counterfactual data, $\text{train}$ and $\text{test}$ represent train and test split, repectively. $\cap$ represents the intersection.
> ||$\text{F}\_\text{train}$|$\text{CF}\_\text{test}$|$\text{F}\_\text{train}$ $\cap$ $\text{CF}\_\text{test}$|
> |:--:|:--:|:--:|:--:|
> |Number|6631|4066|80|
>
> [1] Context is Key: A Benchmark for Forecasting with Essential Textual Information. ICML 2025.
>
> [2] Beyond Naïve Prompting: Strategies for Improved Zero-shot Context-aided Forecasting with LLMs. 2025.
>
> [3] Time-Series Forecasting for Out-of-Distribution Generalization Using Invariant Learning. ICML 2024.
>
> [4] Causal Transformer for Estimating Counterfactual Outcomes. ICML 2022.

---

### Official Review · Reviewer_kMJW · 2025-10-31

**Soundness:** 2
**Presentation:** 3
**Contribution:** 3
**Rating:** 4
**Confidence:** 2

**Summary:**

This paper introduces the task of counterfactual time series forecasting conditioned on unstructured textual descriptions of future events. This addresses limitations of existing methods that rely primarily on historical data or structured future conditions. The proposed TADIFF first attempts to disentangle immutable "intrinsic features" from historical sequences using a diffusion inversion process conditioned on historical text. Then, these features initialize a denoising process conditioned on future text. The proposed DTTC metric enables evaluation in the absence of ground truth, and the empirical results of TADIFF are strong across multiple datasets.

**Strengths:**

1. The paper introduces the task of counterfactual time series forecasting conditioned on unstructured text, moving beyond prior work that often relies on structured or categorical interventions.

2. The TADIFF framework introduces a novel approach within a diffusion model framework designed to handle potential conflicts between historical patterns and future conditions by attempting to separate intrinsic features from external influences.

3. The introduction of the DTTC metric addresses the critical challenge of evaluating forecasts when counterfactual ground truth is unavailable.

4. The proposed method demonstrates effectiveness across various datasets against a comprehensive set of unimodal and multimodal baselines.

**Weaknesses:**

**1. (Major) Ambiguous Intrinsic-features:**
It is conceptually unclear why adding estimated "condition-related noise" via the inverse transition $\psi_t^{-1}$ would result in a representation ($x_{h,T}$) free of $c_h$'s influence. In the DDIM framework [1], $x_{h,T}$ is the specific noise realization that leads back to $x_{h,0}$ when conditioned on $c_h$, this does not guarantee independence from $c_h$. Also there is no rigorous proof or direct empirical validation provided to confirm that $x_{h,T}$ is truly independent of $c_h$. The validation is indirect, relying on ablation studies and t-sne plots (Fig.3).

**2. (Major) Over-reliance on the DTTC Metric:**
The DTTC metric is used for evaluating counterfactual performance and optimizing the model during finetuning. DTTC is used both as an evaluation metric and as the optimization target during counterfactual finetuning , i.e., the model is directly trained to increase DTTC scores. Optimizing against a learned, potentially flawed metric risks overfitting to the metric itself rather than genuinely improving the quality of the counterfactual forecasts
The performance of the DTTC metric varies significantly. For example, the DTTC-I retrieval accuracy on ETTm1 is only 55.06% (Tab.4). This low accuracy casts doubt on the model's ability to reliably capture intrinsic features across all domains.

**3. (Major) Clarity and Consistency:**
In the Sec 3.3, the input $x_t$ is defined such that the historical part $x_{h,0}$ remains clear during training, and a mask $m$ ensures only future components are supervised in Eq.7. However, the inference stage applies the denoising transitions $\psi_t$ to the concatenated sequence $x_{h,0} + x_{f,T}$ (Eq.6). The standard DDIM denoising step updates the entire input sequence. This implies that the known historical part $x_{h,0}$ would be altered during inference, which seems to be inconsistent with the training setup.

**4. (Minor) Potential Bias on the Constructed Text:**
For three real datasets, the “text” is generated from features (tsfresh + templates) rather than organically sourced, which can bias both training and the DTTC encoders toward these templated attributes.


Clarifications are welcome and I will reconsider and raise the score if the questions are addressed.

References:
[1] Song, Jiaming, Chenlin Meng, and Stefano Ermon. "Denoising diffusion implicit models." arXiv preprint arXiv:2010.02502 (2020).

**Questions:**

1. The main supports for independence are Table 3 and a t-SNE plot, which show usefulness but not actual disentanglement/independence. Have the authors tried a simple method (e.g., a classifier that predicts $c_h$​ from $x_{h,T}$)?

---

> ### Author Response · Authors · 2025-11-21
> **Rebuttal by Authors (part 1 of 5)**
>
> > **W1.1** Ambiguous Intrinsic-features: It is conceptually unclear why adding estimated "condition-related noise" via the inverse transition $\psi\_t^{-1}$ would result in a representation ($x\_{h,T}$) free of $c\_h$'s influence. In the DDIM framework [1], $x\_{h,T}$ is the specific noise realization that leads back to $x\_{h,0}$ when conditioned on $c\_h$, this does not guarantee independence from $c\_h$.
>
> Thanks for your suggestions and questions.
>
> The principle of noise estimation in diffusion model supports the independence between $x\_{h,T}$ and $c\_h$. For the denoising process in the standard conditional diffusion model $\hat{x}\_{t-1}=\psi\_t(\hat{x}\_t,c)=\sqrt{\alpha\_{t-1}}\hat{x}\_0+\sqrt{1-\alpha\_{t-1}}\epsilon\_\theta(\hat{x}\_t,c,t)$ (Eq.3 in the paper), the network $\epsilon\_\theta$ estimates the noise component associated with the condition $c\_h$ and removes it from $\hat{x}\_t$. Consequently, the updated sample $\hat{x}\_{t-1}$ becomes more dependent on $c\_h$ than $\hat{x}\_t$ due to the denoising transformation $\psi\_t$. This process can be inverted into $\psi\_t^{-1}$, which adds the condition-related noise back to $\hat{x}\_t$. As a result, the produced sample $\hat{x}\_{t+1}$ becomes less dependent on $c\_h$ than $\hat{x}\_t$.
>
> Several related works [1,2,3] in image editing also incorporate conditions into the forward diffusion process. They show that this operation produces a highly editable representation of clean data, thereby improving editing performance. These findings are consistent with and supportive of our design.
>
> [1] Null-text Inversion for Editing Real Images using Guided Diffusion Models. CVPR 2023.
>
> [2] Prompt-to-Prompt Image Editing with Cross Attention Control. ICLR 2023.
>
> [3] Zero-shot Image-to-Image Translation. SIGGRAPH 2023.
>
> > **W1.2** Also there is no rigorous proof or direct empirical validation provided to confirm that $x\_{h,t}$ is truly independent of $c\_h$. The validation is indirect, relying on ablation studies and t-sne plots (Fig.3).
>
> Thanks for your suggestions and questions.
>
> Recall that we performed t-SNE visualization on the space of initial noise for the inverse diffusion stage for forecasting, with and without text attribution (incorporating conditions in the diffusion forward process) stage (Fig. 3 in paper), and we conducted ablation studies on it (Tab.3 in paper). The results demonstrate that this design significantly affects the initial noise distribution and further improves both factual and counterfactual forecasting.
>
> To provide more direct validation of the dependency between $x\_{h,T}$ and $c\_h$, we train a contrastive learning model (similar to CLIP [4]) to verify whether the relationship between $x\_{h,T}$ and $c\_h$ can be effectively captured (i.e., $x\_{h,T}$ is dependent of $c\_h$) or not ($x\_{h,T}$ is truly independent of $c\_h$).
> - Experiment setting: We first use our model (TADiff) to estimate the initial noise $x\_{h,T}$ based on $x\_{h,0}$ and $c\_h$. Then, we train CLIP models to learn a shared latent space between $x\_{h,T}$ and $c\_h$ through contrastive learning. We evaluate the CLIP model by measuring its accuracy in retrieving the most similar condition $c\_h$ from three candidates given $x\_{h,T}$. For comparison, we also train additional CLIP models between $x\_{h,0}$ and $c\_h$ using the same model architecture.
> - Experimental Results: The results, presented in the following Table 1, prove the independence between $x\_{h,T}$ and $c\_h$. It can be observed that retrieving $c\_h$ given $x\_{h,T}$ yields significantly lower accuracy compared to retrieving $c\_h$ given $x\_{h,0}$, with the performance of the former approaching random guessing. It illustrates that the derived initial noise from our text attribution stage has truly become independent of $c\_h$.
>
> #### Table 1. Accuracy of retrieving the most similar $c\_h$ from 3 candidates, the ideal accuracy of random guessing is approximately 33%.
> |Setting|Synth|ETTm1|Exchange|Traffic|Weather|
> |:--:|:--:|:--:|:--:|:--:|:--:|
> |$x\_{h,0}\rightarrow c\_h$|97.42%|93.34%|98.17%|96.75%|77.71%|
> |$x\_{h,T}\rightarrow c\_h$|32.27%|37.84%|42.93%|39.70%|37.18%|
>
> [4] Learning Transferable Visual Models From Natural Language Supervision. ICML 2021.

---

> ### Author Response · Authors · 2025-11-21
> **Rebuttal by Authors (part 2 of 5)**
>
> > **W2.1** Over-reliance on the DTTC Metric: The DTTC metric is used for evaluating counterfactual performance and optimizing the model during finetuning. DTTC is used both as an evaluation metric and as the optimization target during counterfactual finetuning , i.e., the model is directly trained to increase DTTC scores. Optimizing against a learned, potentially flawed metric risks overfitting to the metric itself rather than genuinely improving the quality of the counterfactual forecasts.
>
> Thanks for your suggestions and questions, and we understand your concern about the proposed novel DTTC metrics.
>
> DTTC scores are reliable metrics for both factual and counterfactual forecasting.
> - DTTC scores effectively capture the intrinsic relationships between historical and future time series, as well as the semantic alignment between future series and external conditions. As demonstrated in Finding 7 of the paper, the DTTC model achieves strong retrieval accuracy, indicating that these scores robustly represent the underlying temporal and semantic features.
> - DTTC scores are also compatible with traditional forecasting metrics such as MAE and MSE. In Finding 3, we ablate the fine-tuning of the DTTC metric on counterfactual data. The results show that using DTTC as the optimization objective not only improves DTTC scores but also preserves the model’s MAE/MSE performance on factual data. This further demonstrates the high reliability of DTTC scores in evaluating the plausibility of the forecasts.
>
> Optimizing against model-based or learned metrics (DTTC scores) is reasonable and widely practiced.
> - Once a metric has been demonstrated to be reliable, using it as an optimization objective is justified, as the metric faithfully reflects forecasting quality. In traditional time-series forecasting, for example, MAE and MSE serve both as evaluation metrics and as direct optimization targets.
> - Numerous prior works adopt model-based or learned metrics as optimization objectives. In the vision domain, for instance, [5] uses a learned CLIP model to adapt image generators to different domains. In the NLP domain, [6] employs a learned reward model to optimize instruction-following capabilities.
>
> In fact, DTTC is primarily designed to address the limitation of traditional metrics which fail to evaluate forecasts under counterfactual conditions due to the absence of ground-truth future observations. Moreover, by using DTTC as the optimization objective, models can additionally leverage counterfactual data as training samples, data that cannot be exploited by traditional methods relying on MSE or MAE as their training objectives.
>
> We understand your concerns regarding the newly proposed metrics and optimization objectives. However, our experimental results have demonstrated their reliability, and we hope that these designs can offer a meaningful direction for the utilization and evaluation of counterfactual data.
>
> [5] StyleGAN-NADA: CLIP-Guided Domain Adaptation of Image Generators. Siggrapgh Asia 2021.
>
> [6] Training language models to follow instructions with human feedback. NeurIPS 2022.

---

> ### Author Response · Authors · 2025-11-21
> **Rebuttal by Authors (part 3 of 5)**
>
> >**W2.2** The performance of the DTTC metric varies significantly. For example, the DTTC-I retrieval accuracy on ETTm1 is only 55.06% (Tab.4). This low accuracy casts doubt on the model's ability to reliably capture intrinsic features across all domains.
>
> We first clarify the experimental setting used in Finding 7.
> - The DTTC model is dataset-specific, meaning it is trained on the training split of each dataset and evaluated on the corresponding test split. Only factual data with paired time series and text are used during training.
>
> Given this dataset-specific training protocol, the DTTC performance naturally varies across datasets. Consequently, each dataset yields a different baseline for evaluating the forecasting models under comparison. Such variation does not affect the validity of our analysis, as the DTTC evaluation is always performed within its own domain.
>
> However, under this experimental setup, DTTC demonstrates strong generalization within each dataset.
> - In Finding 7, both DTTC-I and DTTC-E achieve high retrieval accuracy on most datasets, typically above 80%, and exceeding 95% on several of them.
> - Only DTTC-I obtains 55% accuracy on ETTm1, which may be attributed to characteristics of the dataset. ETTm1 is highly periodic and exhibits strong similarity across time series samples, leading to high intrinsic feature similarity among different instances. Nonetheless, an accuracy of 55% remains substantially higher than random guessing.
>
> Moreover, note that, an oracle setting of the DTTC model could also be considered, where the model is trained on the full dataset rather than only the training split. However, *we do not adopt this configuration* in our final evaluation for the following reasons:
> - Training on the full dataset risks overfitting to the factual samples, making it difficult to assess generalization capability.
> - Since the counterfactual setting involves many unseen combinations of historical sequences and future conditions, a DTTC model with strong generalization is more reliable for evaluation.
>
> > **W3** Clarity and Consistency: In the Sec 3.3, the input $x\_t$ is defined such that the historical part $x\_{h,0}$ remains clear during training, and a mask $m$ ensures only future components are supervised in Eq.7. However, the inference stage applies the denoising transitions $\psi\_t$ to the concatenated sequence $x\_{h,0}+x\_{f,T}$ (Eq.6). The standard DDIM denoising step updates the entire input sequence. This implies that the known historical part $x\_{h,0}$ would be altered during inference, which seems to be inconsistent with the training setup.
>
> Thanks for your reminder. We apologize for the confusion caused by Eq.6.
>
> In our formulation, $x\_{h,0}$ serves as the historical condition and remains unchanged throughout all diffusion steps. At each step $t$, we replace the corresponding patch in the output from the previous step $(t+1)$ with the clean historical sequence $x\_{h,0}$. In this way, the behavior during training and inference remains fully consistent.
> We have refined the formulation in the latest version and explicitly emphasized this replacement operation to avoid further confusion.

---

> ### Author Response · Authors · 2025-11-21
> **Rebuttal by Authors (part 4 of 5)**
>
> > **W4** Potential Bias on the Constructed Text: For three real datasets, the “text” is generated from features (tsfresh + templates) rather than organically sourced, which can bias both training and the DTTC encoders toward these templated attributes.
>
> Thanks for your suggestions.
>
> We shared similar concerns when constructing the domain-agnostic text. Therefore, we applied several operations to increase the diversity of the generated captions:
> - We designed multiple templates for each attribute and randomly sampled from them.
> - We randomly shuffled the order of different attribute descriptions.
>
> To provide more direct evidence that our model (TADiff) does not overfit the constructed templates, we further conducted an experiment to show that TADiff truly learns information from the meaningful content of the text rather than relying on the template-specific wording.
> - Experiment setting: We manually classified the tokens appearing in the generated text into two categories: relevant and irrelevant. Relevant tokens contain meaningful semantics related to the time series, while irrelevant tokens mainly serve to maintain valid sentence structure. For example, in the sentence “The trend direction is up.”, the tokens “trend”, “direction”, and “up” are relevant, whereas “The”, “is”, and “.” are irrelevant. We independently masked relevant and irrelevant tokens at varying ratios and analyzed the resulting performance degradation. We conducted the experiment on the factual data of the ETTm1 and Exchange datasets, where both datasets use the generated text.
> - Experiment results: As shown in Tables 2–5 below, masking irrelevant tokens leads to significantly smaller performance drops compared with masking relevant tokens. Notably, for MAE and MSE, masking 70% of the irrelevant tokens still yields performance comparable to masking only 10% of the relevant tokens. These results demonstrate that TADiff indeed learns from the meaningful semantic information in the generated text rather than overfitting to the constructed templates.
> - We have also added visualization results in the latest version of the paper for clearer interpretation. Please refer to Fig.8 for more details. Thanks again for your valuable suggestions and questions.
>
> We also want to emphasize that not all real datasets rely on the generated text. For the Weather dataset, we use human-expert–annotated text, which provides real and reliable sources. The use of generated text is simply one approach we adopt to broaden the range of available datasets.
>
> #### Table 2. Masking relevant tokens on ETTm1 dataset. $\uparrow$ ($\downarrow$) means the higher (lower), the bettter.
> |Relevant mask ratio|$\downarrow$MAE|$\downarrow$MSE|$\uparrow$DTTC-I|$\uparrow$DTTC-E|
> |:--:|:--:|:--:|:--:|:--:|
> |0.0|$0.57_{\pm 0.04}$|$0.76_{\pm 0.10}$|$10.01_{\pm 0.03}$|$146.91_{\pm 1.78}$|
> |0.1|$0.73_{\pm 0.03}$|$1.26_{\pm 0.16}$|$10.03_{\pm 0.02}$|$140.14_{\pm 2.01}$|
> |0.3|$0.95_{\pm 0.01}$|$1.87_{\pm 0.07}$|$9.66_{\pm 0.03}$|$130.44_{\pm 1.42}$|
> |0.5|$1.15_{\pm 0.01}$|$2.54_{\pm 0.01}$|$9.22_{\pm 0.08}$|$121.61_{\pm 0.40}$|
> |0.7|$1.31_{\pm 0.03}$|$3.07_{\pm 0.09}$|$8.93_{\pm 0.19}$|$114.77_{\pm 1.01}$|
>
> #### Table 3. Masking irrelevant tokens on ETTm1 dataset.
> |Irrelevant mask ratio|$\downarrow$MAE|$\downarrow$MSE|$\uparrow$DTTC-I|$\uparrow$DTTC-E|
> |:--:|:--:|:--:|:--:|:--:|
> |0.0|$0.57_{\pm 0.04}$|$0.76_{\pm 0.10}$|$10.01_{\pm 0.03}$|$146.91_{\pm 1.78}$|
> |0.1|$0.62_{\pm 0.04}$|$0.81_{\pm 0.19}$|$10.01_{\pm 0.00}$|$145.07_{\pm 2.17}$|
> |0.3|$0.70_{\pm 0.00}$|$1.08_{\pm 0.06}$|$9.93_{\pm 0.03}$|$140.75_{\pm 2.79}$|
> |0.5|$0.78_{\pm 0.05}$|$1.28_{\pm 0.10}$|$9.88_{\pm 0.07}$|$131.03_{\pm 4.89}$|
> |0.7|$0.82_{\pm 0.08}$|$1.35_{\pm 0.20}$|$9.77_{\pm 0.05}$|$125.54_{\pm 7.03}$|
>
> #### Table 4. Masking relevant tokens on Exchange dataset.
> |Relevant mask ratio|$\downarrow$MAE|$\downarrow$MSE|$\uparrow$DTTC-I|$\uparrow$DTTC-E|
> |:--:|:--:|:--:|:--:|:--:|
> |0.0|$0.15_{\pm 0.01}$|$0.04_{\pm 0.00}$|$17.37_{\pm 0.04}$|$216.03_{\pm 3.11}$|
> |0.1|$0.18_{\pm 0.00}$|$0.07_{\pm 0.01}$|$17.09_{\pm 0.07}$|$192.78_{\pm 0.35}$|
> |0.3|$0.23_{\pm 0.01}$|$0.17_{\pm 0.05}$|$16.84_{\pm 0.07}$|$166.39_{\pm 2.88}$|
> |0.5|$0.33_{\pm 0.03}$|$0.45_{\pm 0.19}$|$16.49_{\pm 0.12}$|$139.75_{\pm 4.90}$|
> |0.7|$0.40_{\pm 0.05}$|$0.92_{\pm 0.55}$|$16.05_{\pm 0.28}$|$122.30_{\pm 8.17}$|
>
> #### Table 5. Masking irrelevant tokens on Exchange dataset.
> |Irrelevant mask ratio|$\downarrow$MAE|$\downarrow$MSE|$\uparrow$DTTC-I|$\uparrow$DTTC-E|
> |:--:|:--:|:--:|:--:|:--:|
> |0.0|$0.15_{\pm 0.01}$|$0.04_{\pm 0.00}$|$17.37_{\pm 0.04}$|$216.03_{\pm 3.11}$|
> |0.1|$0.15_{\pm 0.00}$|$0.05_{\pm 0.00}$|$17.23_{\pm 0.01}$|$214.82_{\pm 0.29}$|
> |0.3|$0.17_{\pm 0.00}$|$0.05_{\pm 0.00}$|$17.15_{\pm 0.05}$|$199.88_{\pm 1.52}$|
> |0.5|$0.19_{\pm 0.00}$|$0.06_{\pm 0.00}$|$17.08_{\pm 0.05}$|$178.61_{\pm 2.79}$|
> |0.7|$0.19_{\pm 0.00}$|$0.07_{\pm 0.00}$|$16.98_{\pm 0.03}$|$158.59_{\pm 3.52}$|

---

> ### Author Response · Authors · 2025-11-21
> **Rebuttal by Authors (part 5 of 5)**
>
> > **Q1** The main supports for independence are Table 3 and a t-SNE plot, which show usefulness but not actual disentanglement/independence. Have the authors tried a simple method (e.g., a classifier that predicts $c\_h$ from $x\_{h,T}$)?
>
> Thank you for your comments and valuable suggestions!
>
> As mentioned in the response of **W1**, per your suggestions, we empirically prove that $x\_{h,T}$ is independent with $c\_h$. In summary, we train a dataset-specific CLIP model (which plays a role similar to a classifier) for each dataset, compare the $c\_h$ retrieving accuracy between given clear data $x\_{h,0}$ and noisy data $x\_{h,T}$. The results show that retrieving $c\_h$ given $x\_{h,T}$ yields significantly lower accuracy compared to retrieving $c\_h$ given $x\_{h,0}$, with the performance of the former approaching random guessing. We believe this experiment provides direct evidence supporting the independence.

---

### Author Response · Authors · 2025-11-21
**General Responses**

We thank all the reviewers for their valuable feedback.

We are encouraged by the positive comments from the reviewers, such as:
- The proposed text-conditioned counterfactual time series forecasting task is interesting, valuable, and has broad real-world applications (Reviewers kMJW, iGqX, FZhN).
- The proposed text-attributive forecasting model TADiff and the training strategy on counterfactual data are novel and technically solid (Reviewers kMJW, Tfgp, iGqX, FZhN).
- The proposed evaluation metrics (DTTC-I and DTTC-E) for counterfactual settings are novel and meaningful (Reviewers kMJW, Tfgp, iGqX).
- The experiments demonstrate the effectiveness of TADiff and offer many useful insights (Reviewers kMJW, Tfgp).
- The paper is well presented with clear problem motivation and setup (Reviewer FZhN).

The reviewers' suggestions have greatly helped us improve our work. We have made corresponding revisions based on their feedback (highlighted in blue in the revised paper), summarized as follows:
- Added more details on dataset construction, experimental settings, and analysis, as suggested by reviewers Tfgp, iGqX, and FZhN.
- Expanded the related work section and further clarified our position and contributions, as suggested by reviewers Tfgp and FZhN.
- Added discussions and experimental analyses regarding the potential risks of constructed text templates, as suggested by reviewers kMJW and Tfgp.
- Improved wording and figure presentation throughout the paper, as suggested by reviewers kMJW, Tfgp, and FZhN.

---

### Meta-Review · Area_Chair_ECuG · 2026-01-07

**Summary:**

This paper proposes a diffusion-based method for text-conditioned time series forecasting.

I appreciate the tremendous efforts made by the authors during the rebuttal phase. Most of the reviewers are still negative towards this manuscript. After carefully reading the revised manuscript and the reviews, I have to recommend rejection due to the following concerns.

(1) Although the paper is framed as a counterfactual forecasting problem, the setting is closer to a time series generation task (what-if task) rather than forecasting in the classical sense. Since no ground-truth data exist for the counterfactual scenarios, the effectiveness of the proposed model cannot be fully validated. It is also difficult to judge whether the proposed metrics are appropriate.

Related to this point, if the task is indeed generation-oriented, the paper should be more clearly contextualized within the multimodal time series generation literature. In that case, relevant generative models should be included as primary baselines. The current comparisons are mostly against forecasting methods, which do not align well with the underlying problem formulation.

(2) If the task is defined as forecasting, a key assumption—that future textual conditions are available at prediction time—is not realistic in most real-world scenarios. Since future conditions are inherently unknown, this further weakens the practical validity of the proposed setting.

(3) The title uses counterfactual forecasting, while the paper and experiments cover both factual forecasting and counterfactual generation. In its current form, the title does not accurately summarize the scope or focus of the work, which may be misleading to readers.

(4) Several key components of the methodology are not clearly explained in the main body, which affects the overall readability.

In summary, while the direction explored in this paper is potentially interesting, the remaining issues regarding problem definition, evaluation validity, realism, and presentation are substantial. Therefore, I recommend rejection.

**Reviewer Concerns:**

Some concerns—such as clarification of the proposed method and parts of the evaluation—have been addressed in the rebuttal. However, the core issues regarding the problem setting and its contextualization within the existing research landscape remain largely unresolved.

**Reviewer Scores:**

Reviewer Tfgp has raised his/her score from 4 to 6, while the other three reviewers would keep their initial scores as 4, 4, 4.

---

### Decision · Program_Chairs · 2026-01-26

Reject